



# Measurements of atmospheric He/N₂ as an indicator of fossil fuel extraction and stratospheric circulation

Benjamin Birner[1], William Paplawsky[1], Jeffrey Severinghaus[1], and Ralph F. Keeling[1]

[1]Scripps Institution of Oceanography, UC San Diego, La Jolla, CA 92093, USA

*Correspondence to*: Benjamin Birner (bbirner@ucsd.edu)

**Abstract.** The atmospheric He/N₂ ratio is expected to be increasing due to the emission of He associated with fossil fuels and is expected to also vary in both space and time due to gravitational separation in the stratosphere. These signals may be useful indicators of fossil-fuel exploitation and variability in stratospheric circulation, but direct measurements of He/N₂ ratio are lacking on all time scales. Here we present a high-precision custom inlet system for mass spectrometers that continuously stabilizes the flow of gas during sample-standard comparison and removes all non-noble gases from the gas stream, enabling unprecedented accuracy in measurement of relative changes in the $^4$He/N₂ ratio. Repeat measurements of the same combination of high-pressure tanks using our inlet system achieves a reproducibility of ~10 per meg (i.e. 0.001%) in 6–8h analyses. This compares to interannual changes of He/N₂ gravitational enrichment at ~35 km in the mid latitude stratosphere of order 300–400 per meg, and an annual tropospheric increase from human fossil fuel activity of less than  ~30 per meg y$^{-1}$ (bounded by previous work on helium isotopes). The gettering and flow-stabilizing inlet may also be used for the analysis of other noble gas isotopes and could resolve previously unobserved seasonal cycles in Kr/N₂ and Xe/N₂.

## 1 Introduction

The atmospheric mole fraction of helium in dry air is typical ~5.24 ppm (Glückauf, 1944) with an isotopic abundance of $^4$He about $10^6$ times greater than $^3$He. On geological time scales, the natural concentration of $^4$He in the atmosphere is set by a balance of $^4$He loss to space and $^4$He release from the Earth's crust, where it is produced by radioactive decay of uranium and thorium (Kockarts, 1973; Pierson-Wickmann et al., 2001; Sano et al., 2013; Torgersen, 1989; Zartman et al., 1961). Over the past century, human exploitation of fossil fuels likely has accelerated the release of crustal He, but the observational evidence of a secular increase of atmospheric $^4$He remains ambiguous presumably due to a lack of analytical precision (Boucher et al., 2018c; Lupton and Evans, 2013, 2004; Mabry et al., 2015; Oliver et al., 1984; Pierson-Wickmann et al., 2001; Sano et al., 1989). Additionally, recent measurements and model simulations reveal a small depletion of gases heavier than air in the stratosphere by gravitational separation (Belikov et al., 2019; Birner et al., 2020; Ishidoya et al., 2020, 2018, 2013, 2008; Sugawara et al., 2018) suggesting a corresponding enrichment of the light gas helium. Gravitational separation is only partially counteracted by the large-scale stratospheric circulation and mixing, which tends to homogenize the atmosphere. Variability



in stratospheric circulation and stratosphere-troposphere exchange (STE) could therefore impact the degree of fractionation and cause additional interannual changes in the stratospheric and, to a much lesser extent, the tropospheric abundance of $^4$He. Measurements of He/N$_2$ may provide an alternative indicator of variations in stratospheric circulation and STE. An improved understanding of STE is critical because stratospheric circulation changes affect tropospheric trends of societally-important trace and greenhouse gases such as N$_2$O, CH$_4$, $^{14}$C, O$_3$ and CFCs (Arblaster et al., 2014; Graven et al., 2012; Hamilton and

Fan, 2000; Hegglin and Shepherd, 2009; Montzka et al., 2018; Nevison et al., 2011; Simmonds et al., 2013). These gases all have significant sources or sinks in the stratosphere that cause strong stratosphere-troposphere concentration differences. Global circulation models consistently predict an acceleration of the stratospheric Brewer-Dobson Circulation (BDC; Brewer, 1949; Dobson, 1956) under global warming (Butchart, 2014). Stratospheric circulation is also naturally modulated on a range of shorter time scales from synoptic-scale events to decadal variations (e.g., Holton et al., 1995; Li et al., 2012; Flury et al.,

2013; Butchart, 2014; Ray et al., 2014). Circulation changes have typically been observed using measurements of numerous different trace gases in the stratosphere (e.g., CO$_2$, SF$_6$, H$_2$O, O$_3$, CO, or N$_2$O) (e.g., Bönisch et al., 2009; Engel et al., 2009, 2017; Ray et al., 2010; Haenel et al., 2015). However, interpretation of these tracers of stratospheric circulation is complicated by complex chemical source and sink processes, whereas gravitational fractionation of He/N$_2$ is governed by comparatively simple physics.

Atmospheric He/N$_2$ measurements may also provide an indication of the history of fossil-fuel usage. Previous attempts to measure the fossil-fuel signal in He has centered on measurements of changes in the atmospheric $^3$He/$^4$He isotope ratio (Boucher et al., 2018c; Lupton and Evans, 2013, 2004; Mabry et al., 2015; Oliver et al., 1984; Sano et al., 1989). However, measurements of the $^3$He/$^4$He ratio are fundamentally limited by the extremely low abundance of $^3$He (e.g., Mabry et al., 2015; Boucher et al., 2018b), with only 1 in 730,000 He atoms being $^3$He. Therefore, the precision on individual $^3$He/$^4$He analyses is

limited to ~±0.2%. This is insufficient for the detection of the stratospheric and anthropogenic signals we are interested in, and which we estimate to cause variations in the atmospheric $^4$He mole fraction on the order of 0.0030 to 0.04% y$^{-1}$ (see section 2.1 & 2.2.). Moreover, small changes in $^3$He from radioactive decay of tritium in nuclear warheads may complicate the interpretation of $^3$He/$^4$He results (e.g., Christine Boucher et al., 2018c; Lupton and Evans, 2004).

Here we describe a method to measure relative differences in $^4$He mole fraction ($^4$He/M) between two large samples of air

using a custom mass spectrometer inlet system. The helium mole fraction can later be mathematically translated to our target ratio, $^4$He/N$_2$, given supplementary measurements of O$_2$, Ar, and CO$_2$ (see discussion). This is advantageous because N$_2$ is near-constant in the atmosphere making $^4$He/N$_2$ more readily interpretable than $^4$He/M. The $^4$He/M method depends on stabilization of the gas flow to the ion source between a sample and standard gas to achieve high precision differencing. Novel elements in our setup include continuous flow removal of reactive gases via titanium gettering immediately upstream of the

mass spectrometer inlet, and the use of an actively-controlled open split (Henneberg et al., 1975) for balancing pressures upstream of a shared capillary directed towards the mass spectrometer. Gas handling techniques, the inlet system and the continuous-flow getter oven are described in detail below.



## 1.1 Gravitational fractionation of He/N₂ in the stratosphere

The notion that the stratospheric and tropospheric He/$N_2$ ratio must vary in response to fluctuations in stratospheric circulation
is based on studies of the atmospheric Ar/$N_2$ ratio (Birner et al., 2020; Ishidoya et al., 2020). Relative changes in the Ar/$N_2$
ratio (or He/$N_2$) are commonly expressed in delta notation:

$$\delta(Ar/N_2) = \frac{\left(\frac{Ar}{N_2}\right)_{SA}}{\left(\frac{Ar}{N_2}\right)_{ST}} - 1 \tag{1}$$

where subscripts SA and ST refer to the ratio in a sample and a reference gas mixture, respectively. $\delta(Ar/N_2)$ is multiplied by
$10^6$ and expressed in "per meg" units.

Sensitivity tests with the 2-D chemical-dynamical-radiative model of the atmosphere SOCRATES by Ishidoya et al. (2020)
indicate that significant temporal changes in stratospheric Ar/$N_2$ should occur in response to an acceleration or deceleration of
the BDC. The simulations also suggest a weak stratospheric influence on tropospheric Ar/$N_2$. Ishidoya et al find that imposing
a gradual acceleration of the BDC of 4% dec⁻¹ leads to a 40 per meg dec⁻¹ increase in $\delta(Ar/N_2)$ at ~35 km altitude in northern
mid-latitudes, and a corresponding 1.3 per meg dec⁻¹ decrease of $\delta(Ar/N_2)$ in the troposphere. Furthermore, they find that
imposing 3-year periodic changes of 10% in BDC yields anomalies of ±25 and ±0.4 per meg in stratospheric and tropospheric
$\delta(Ar/N_2)$, respectively. Tropospheric observations of $\delta(Ar/N_2)$ by Ishidoya et al. (2020) would be consistent with larger STE-
induced interannual changes of tropospheric Ar/$N_2$. Variability of the BDC on the order of 10% or more on seasonal to decadal
time scales is consistent with published estimates (Flury et al., 2013; Ray et al., 2014; Salby and Callaghan, 2006).

The atmospheric He/$N_2$ ratio must be more strongly impacted by gravitational fractionation than Ar/$N_2$ due to the larger mass
difference and higher diffusivity of He than Ar, which brings He closer to gravitational equilibrium. The gravitational
fractionation effect on He/$N_2$ can be scaled from Ar/$N_2$ (Birner et al., 2020) using the molar mass difference to air $\Delta M_i$ ($\Delta M_i = M_i - 0.02896$ kg mol⁻¹) and the molecular diffusivity $D_i$ of gas $i$ in air as:

$$\delta(\text{He/N}_2) = \frac{\left(\frac{He}{N_2}\right)_{SA}}{\left(\frac{He}{N_2}\right)_{ST}} \approx \frac{D_{He}^{air}\Delta M_{He} - D_{N2}^{air}\Delta M_{N2}}{D_{Ar}^{air}\Delta M_{Ar} - D_{N2}^{air}\Delta M_{N2}}\delta(Ar/N_2). \tag{2}$$

Using the method of Fuller et al. as reported in (Reid et al., 1987), $D_{He}$ is 3.6 times greater than $D_{Ar}$ and $\Delta M$ is more than twice
as large, making $\delta(\text{He/N}_2)$ ~7.5 times more strongly fractionated by gravity than $\delta(Ar/N_2)$ in the stratosphere.

## 1.2 Other controls on tropospheric He/N₂

A variety of known natural processes influence tropospheric $^4$He/$N_2$ is summarized in Figure 1 and Table 1. Natural $^4$He release
from the Earth's crust is mediated by volcanism, ground water discharge and diffusive leakage. At the same time, helium is


lost to space by thermal and non-thermal escape (Kockarts, 1973; Oliver et al., 1984; Pierson-Wickmann et al., 2001; Sano et al., 2013; Torgersen, 1989). Based on these natural fluxes and the total atmospheric burden, the atmospheric residence time of $^4$He is estimated to be ~1 million years.

Over the few last centuries, He release from fossil-fuel extraction has dwarfed the natural release rates of $^4$He by several orders of magnitude. Based on knowledge of fossil fuel usage and He content of the material (Table 1) the additional $^4$He release rate is estimated to be of order 3 to $30 \times 10^{10}$ mole yr$^{-1}$ (e.g., Oliver et al., 1984; Sano et al., 1989, 2013; Pierson-Wickmann et al., 2001) implying that $^3$He/$^4$He should be decreasing at rates between 35 and 350 per meg y$^{-1}$. However, in contrast to these predictions and some earlier observations (Oliver et al., 1984; Pierson-Wickmann et al., 2001; Sano et al., 2010, 1989), no

significant trend in atmospheric $^3$He/$^4$He has been observed using archived air samples spanning from the beginning of the 20[th] century to today at the level of roughly ±30 per meg per year, suggesting similarly small increase rates in $\delta(He/N_2)$ (Boucher et al., 2018c; Lupton and Evans, 2013, 2004; Mabry et al., 2015).

He release from fossil-fuel extraction is also expected to impose an interhemispheric gradient in $\delta(He/N_2)$. A rough upper bound can be estimated by assuming all fossil-fuel derived He emissions occur in the Northern Hemisphere and

interhemispheric mixing of the atmosphere has a time scale of about one year. This would yield a north-south difference of 30 per meg, equal to the expected annual rise in $\delta(He/N_2)$.

Seasonal and long-term ocean warming can cause small changes in He/N$_2$, mainly due to the impact on N$_2$. From observations of $\delta(Ar/N_2)$ (Keeling et al., 2004) and solubility data of Ar, He and N$_2$ (Hamme and Emerson, 2004; Weiss, 1971), we estimate that the impact on He/N$_2$ of air-sea exchanges is on the order of 0.16 per meg y$^{-1}$ for the secular ocean warming trend and 3-9

per meg for seasonal heat exchanges. Therefore, the ratio of stratospheric signals to ocean warming is ~12 times greater for He/N$_2$ than Ar/N$_2$ and the effect of slow ocean warming is over two orders of magnitude smaller than the influence of fossil fuel exploitation.

The He/N$_2$ ratio could also be impacted by processes changing atmospheric N$_2$. However, the annual removal of $7.5 \times 10^{12}$ moles N$_2$ y$^{-1}$ by anthropogenic nitrogen fixation in agriculture, combustion, and industry is clearly negligible compared to the

~$1.4 \times 10^{20}$ moles of N$_2$ in the whole atmosphere (Fowler et al., 2013). Volcanic emissions of N$_2$ are likewise negligible, order $10^9$ moles y$^{-1}$.

## 2 Methods

Our He/N$_2$ analysis method relies on measuring the helium mole fraction relative difference between an air sample and a standard reference gas using a single collector for $^4$He$^+$ on a magnetic sector mass spectrometer (MS). Crucially, whole dry air

is pressure-stabilized to a high level prior to gettering, so that the beam intensity ratio being measured is effectively the $^4$He to air ratio. Measurements of the He mole fraction difference can also be expressed similarly to Eq. (1) as $\delta(^4He/M)$ where M is total moles. By applying small corrections for variations in O$_2$/N$_2$, Ar/N$_2$, and CO$_2$, the quantity $\delta(^4He/M)$ is easily related to $\delta(^4He/N_2)$.



The MS is interfaced to a custom inlet system with on-line gettering and active flow stabilization using an actively pressure-

controlled open split (Henneberg et al., 1975), as shown in Figure 2.

## 2.1   The Inlet system

The design of the inlet system incorporates elements of an open split (Henneberg et al., 1975) but further stabilizes the pressure using active control elements and allows active switching between a sample (SA) and reference standard gas stream (ST). Pneumatically-actuated pistons (Fig. 2C) alternately slide 0.3-mm tubes exhausting sample or standard gas close to a shared

intake capillary which is placed at the end of the stabilization chamber (Fig. 2D) and connects the chamber to the getter oven and MS. The chamber is shaped as a funnel to guide the sliding tubing into a reproducible resting position. Variations in chamber pressure are measured with a 0.2-Torr MKS 223B differential pressure gauge and are limited to better than 1 part in $10^6$ by opening an MKS Type 248 Control Valve which allows most of the gas in the stabilization chamber to be pumped away by a vacuum pump. The valve is controlled via an MKS 250E Control Module. The 0.3mm tubes enter the chamber through

sliding seals lubricated with vacuum grease, thus allowing the chamber to be operated at a selected pressure above or below ambient. The default setting is 14 psia (96.5 kPa). The shared outlet capillary is crimped and thermally insulated. The pressure in the getter oven (Fig. 2E) is about 2 mTorr (0.3 Pa) because the getter material effectively acts as a vacuum pump.

## 2.2   Continuous-flow gettering

In the getter chamber (Fig. 2E), high purity titanium sponge (Ti) irreversibly reacts with $N_2$, $O_2$, $CO_2$, and other non-noble

gases in air to form titanium nitride (TiN), titanium dioxide ($TiO_2$), titanium carbide (TiC) and other compounds at ~850°C. This increases the concentration of He in the gas mixture by a factor of about 100, boosting precision. The getter oven is manufactured from heat resistant stainless steel (SS310) and equipped with VCR face seals for easy maintenance. The temperature of the getter oven is determined by manually adjusting the power provided to the furnace surrounding the getter. The furnace is additionally equipped with an independent limit controller for safety.

The gettering efficiency depends on the furnace's temperature and must be balanced against material tolerance and increased evolution of $H_2$ gas from the metal in the getter. $H_2$ forms a solid solution in Ti and is continuously released to the gas stream when Ti is heated. The solution process is reversible and $H_2$ is absorbed if the Ti is cooled down. $H_2$ could interact with $He^+$ in the source or combine with ionized gas to form hydride compounds such as $ArH^+$ (Fig. 3) However, since the $H_2$ flux into the gas stream varies slowly compared to the 30-second switching timescale, the impact of $H^+$ cancels during sample-standard

comparison. In its current size (~10–12 g Ti), the getter can be used for 70–80 h before the Ti must be replaced. This requires breaking vacuum in the inlet approximately once every four weeks depending on usage. After replacement, fresh titanium is gradually heated to 900°C over ~12h in isolation from the MS to allow degassing without contaminating the MS. A coarse mesh of metal wire and 2 μm SWAGELOK filters on both sides of the getter prevent getter-derived dust from entering the vacuum system and MS.





### 2.3    Inlet operation

We have developed customized scripts using the software ISODAT provided with any MAT253 mass spectrometer to control the inlet system and operate the pneumatic actuators for He/M analysis (Fig. 3). In a typical run, the instrument performs sample-standard gas switching with a ~30 second switching time (~60 sec full cycle), using a conservative 18-second idle time to ensure complete flush-out of the stabilization chamber and the getter oven. As customary in dynamic MS noble gas application, we group each analysis into blocks consisting of (i) adjusting the accelerating voltage to find the center of the $^4$He peak followed by (ii) 20 sample-standard comparisons. Background concentrations of $^4$He in the MS are determined daily and subtracted. Data are quality controlled and anomalous cycles are rejected when delta values deviate by more than 3 standard deviations from the smoothed time series or when there are abrupt changes detected in the ion beam associated with instability in the MS source (not shown). ISODAT also monitors the MS source pressure and closes the external change-over-valve (Fig. 2) to protect the MS in case of a pressure control failure.

### 2.4    Gas handling and sample delivery systems

Air can be delivered to the inlet system from either a pair of high-pressure gas cylinders (Fig. 2A) or from a continuous-flow system (Fig. 2B) pumping locally-sampled ambient air directly to the laboratory. For He/$N_2$ reference material, we rely on compressed dry air stored in high pressure cylinders, as is conventional for atmospheric measurements of $O_2$/$N_2$, $CO_2$, and Ar/$N_2$ (Keeling et al., 2007). All cylinders are stored horizontally for 2 days in a thermal enclosure before analysis to minimize the risk of thermal fractionation. The pressure is dropped to slightly above ambient directly at the head valve of high-pressure cylinders using capillaries rather than regulators. The use of capillaries ensures that all wetted parts are exclusively metal, which is impermeable to He, and eliminates problems we encountered using regulators during initial tests. Due to the use of capillaries, the gas delivery system cannot be evacuated efficiently and instead must be purged for several hours ahead of analysis until the signal stabilizes. The flow rates in the lines are monitored using 0.1l Omron DF6-P flow meters and are manually balanced at around 27-28 cm$^3$ min$^{-1}$ before every analysis by adjusting the crimping of the capillaries. Sample and standard gas streams each flow through a -80°C cold trap made from 1/4" stainless steel tubing before entering the pressure stabilization chamber (Fig. 2D).

## 3  Results

### 3.1    Gettering performance

A mass scan of medical air introduced through the gettering and flow-stabilizing inlet system revealed that $N_2$ and $O_2$ are almost completely removed from the air by the on-line getter (Fig. 4). $^{40}$Ar ions with one or more charges yield the largest beams in the scan followed by $^{36}$Ar, and $H_2$ evolving from the hot metal in the getter oven.



## 3.2    Response time

Our setup demonstrates the ability to transition between sample and standard gas with a e-folding time scale of ~4 seconds
(Fig. 5). The e-folding time is primarily controlled by the volume of the getter and the total flow of gas through the getter.
Regions of the inlet system upstream of the getter experience ~100x faster flushing than downstream of the getter because the
gas upstream still contains $N_2$ and $O_2$ and hence flows much faster. The e-folding time does not change substantially over the
life span of the getter.

## 3.3    Analytical precision

Using the default 60 sec sample-standard cycle, the gettering and flow-stabilizing inlet system achieves an internal precision
in $\delta(He/M)$ of approximately ±15 per meg over 1.5h and ±8 per meg (1σ) for samples run 6h or longer (Figs. 6&7). The
reproducibility of repeated 6–8h measurements of the same sample and standard gas cylinder combination is comparable and
essentially as expected from the shot-noise on the $^4$He ion current.

The zero enrichment, i.e., the delta value observed when introducing the same gas through sample and standard side of the
inlet, is generally small and stable over time. It is tested by mounting the crimped delivery capillaries (Fig. 2A) to a tee fitting,
which splits the gas stream at high pressure from a single tank of air. This tee minimizes thermal fractionation by dividing the
flow at a junction machined into the center of a large brass block (Keeling, 1988). Identical delta values (within error) obtained
after reversing the outlet from the tee demonstrate that no measurable fractionation occurs within the tee and therefore that the

zero enrichment reflects a persistent asymmetry somewhere downstream, most likely within the open split. The typical zero
enrichment varies slightly with the mean flow of gas into the stabilization chamber (F), the pressure in the chamber (P), and
the flow offset between SA and ST side (ΔF) before entering the stabilization chamber (Fig. 7). Weighted multiple linear
regression analysis using 3 different pressure levels (9 psi, 14 psi, and 16 psi, i.e, 62.1, 96.5, and 110.3 kPa) reveals that the
zero enrichment value decreases by 2.8±0.9 per meg per 1 $cm^3$ $min^{-1}$ change in mean flow away from 27.5 $cm^3$ $min^{-1}$ and

increases by 17.2±4.8 per meg per 1 $cm^3$ $min^{-1}$ flow imbalance between SA and ST. The dependence of $\delta(He/M)$ on F and ΔF
is significant at the 5% level. For a balanced flow of 27.5 $cm^3$ $min^{-1}$ at 9, 14 and 16 psi pressure in the stabilization chamber,
the mean zero enrichment is -9.61±7.2 per meg, 1±3.7 per meg and -15.7±4.7 per meg, respectively. P is held constant for
measurements at 14 psi. F and ΔF are stable over 8h to within ±0.2 $cm^3$ $min^{-1}$. This typically yields a correction for mean gas
flow and flow imbalance of less than 10 per meg with an uncertainty smaller than 6 per meg, which increases the overall

analytical uncertainty in repeat tank analysis from 8 to 10 per meg.



## 4 Discussion

The gettering and flow-stabilizing inlet system has demonstrated the ability to determine the helium mole fraction difference between a sample and standard gas, $\delta(He/M)$, to about 10 per meg in a single 6–8h analysis. $\delta(He/M)$ can be related to $\delta(He/N_2)$ using

$$\delta(He/N_2) \simeq \delta(He/M) + \delta(O_2/N_2)X_{O_2} + \delta(Ar/N_2)X_{Ar} + dX_{CO_2} \tag{1}$$

as derived in Box 1, using independent measurements of $\delta(O_2/N_2)$, $\delta(Ar/N_2)$, and $dX_{CO2}$ (Keeling et al., 2004, 1998). These corrections are relatively small and therefore do not significantly contribute to the overall uncertainty of $\delta(He/N_2)$. The long-term atmospheric changes in $\delta(O_2/N_2)$ ~ -19 per meg yr$^{-1}$ and $dX_{CO2}$ ~2.5 ppm yr$^{-1}$ yield corrections of approximately -4 per meg yr$^{-1}$ and +2.5 per meg yr$^{-1}$, respectively. The seasonal variations in $\delta(O_2/N_2)X_{O_2}$ and $dX_{CO2}$ partly cancel, yielding net seasonal corrections of ~10 per meg in both hemispheres. The term $\delta(Ar/N_2)X_{Ar}$ contributes variations less than 1 per meg on 215 all time scales.

---

**Box 1. Deriving the helium-to-nitrogen ratio from $\delta(He/M)$**

A relationship between $\delta(He/N_2)$ and $\delta(He/M)$ can be derived from:

$$\delta(He/N_2) = \frac{dHe}{He} - \frac{dN_2}{N_2} = \frac{dHe}{He} - \frac{dM}{M} - \frac{dN_2}{N_2} + \frac{dM}{M} = \delta(He/M) - \frac{dN_2}{N_2} + \frac{dN_2 + dO_2 + dAr + dCO_2}{M}$$

Using $\frac{dN_2}{M} = \frac{dN_2}{N_2} X_{N_2}$, $\frac{dO_2}{M} = \frac{dO_2}{O_2} X_{O_2}$, etc, where $X_i$ is the mole fraction of gas i, yields

$$\delta(He/N_2) = \delta(He/M) + \frac{dN_2}{N_2}\left[-1 + X_{N_2} + X_{O_2} + X_{Ar} + X_{CO_2} + X_{H_2O} \ldots\right] + \left[\frac{dO_2}{O_2} - \frac{dN_2}{N_2}\right]X_{O_2} + \left[\frac{dAr}{Ar} - \frac{dN_2}{N_2}\right]X_{Ar}$$

$$+ \left[\frac{dCO_2}{CO_2} - \frac{dN_2}{N_2}\right]X_{CO_2} + \cdots$$

This can be simplified to Eq. (3) using $\left[\frac{dCO_2}{CO_2} - \frac{dN_2}{N_2}\right]X_{CO_2} = dX_{CO_2}$, which follows because relative changes in $CO_2$ are much larger than relative changes in $N_2$.

---

### 4.1 Potential applications

The presented He/N$_2$ measurement capability has a range of possible applications. Our primary targets are (i) to use stratospheric $\delta(He/N_2)$ as a tracer of the large-scale stratospheric circulation and (ii) to evaluate tropospheric $\delta(He/N_2)$ trends 220 as a possible indicator of anthropogenic fossil fuel exploitation.

We expect an excellent signal-to-noise ratio for the detection of stratospheric changes in $\delta(He/N_2)$. Interannual variability in stratospheric $\delta(He/N_2)$ is likely on the order 300–400 per meg (Table 1). Repeat 6–8h measurements of a high-pressure tank currently achieve a precision of ~10 per meg, or about 40 times better than the stratospheric signal. Associated changes in tropospheric $\delta(He/N_2)$, in contrast, are likely much smaller at around 6 per meg and therefore at or below the current limit of





detection even after averaging of multiple samples. The reproducibility of measurements also depends on adequate calibration strategies. The short-term reproducibility of high-pressure cylinders shown in Figure 6 and the long-term stability established for $O_2/N_2$, $CO_2$, and $Ar/N_2$ standard gases in previous work (Keeling et al., 2007) suggest long-term stability in $\delta(He/N_2)$ is achievable but needs further evaluation.

    $He/N_2$ measurements can help quantify the anthropogenic $^4He$ release over time due to fossil fuel extraction (Boucher et al.,
2018c; Lupton and Evans, 2013, 2004; Mabry et al., 2015; Oliver et al., 1984; Sano et al., 2010, 1989). Although theoretical predictions clearly support an anthropogenic $^4He$ increase, past observational studies produced conflicting evidence. Recent improvements in analytical methods and sampling have narrowed the uncertainty in $^3He/^4He$ trend estimates to <30 per meg y$^{-1}$ with a mean statistically indistinguishable from zero (Table 1). However, with a precision of ~10 per meg on single samples, measurements of $\delta(^4He/N_2)$ on decades-old archived air may allow trend detection to ~1 per meg y$^{-1}$ or better, while also
avoiding possible complications from $^3He$ emissions.

    Another possible application is the investigation of spatial gradients in atmospheric $\delta(He/N_2)$ caused by the distribution of local volcanic or anthropogenic sources (e.g., Sano et al., 2010; Boucher et al., 2018c). High precision $\delta(He/N_2)$ may allow the detection of diffuse helium release in regions of volcanic activity (Boucher et al., 2018b). Furthermore, global north-south $\delta(He/N_2)$ gradients from anthropogenic emission sources concentrated in the Northern Hemisphere are likely on the order of
10s of per meg and thus may also be detectable directly. Alternatively, studies could target more local gradients around oil or natural gas facilities that are likely even greater.

    The method developed here is potentially applicable to measure the abundance of any noble gas in air. The intensity of the ion beam and thus the precision for different noble gases depends on their natural abundance and ionization efficiency in the MS source. $^{20}Ne$ and $^{22}Ne$ have isobaric interferences from doubly charged Ar and $CO_2$, but Kr and Xe yield usable ion beams
(Table 2). We estimate a precision of ~5 and ~19 per meg for repeat 6–8h analyses of $\delta(^{84}Kr/^{28}N_2)$ and $\delta(^{129}Xe/^{28}N_2)$ respectively, by assuming that precision scales with the square root of the total ions counted as expected from shot-noise behavior. This estimate compares favorably to the precision currently reported in conventional dual inlet mass spectrometry studies (Baggenstos et al., 2019; Bereiter et al., 2018). For example, Baggenstos et al. (2019) achieved a precision of 88 per meg and 203 per meg for repeat ~2h analyses of $\delta(^{84}Kr/^{40}Ar)$ and $\delta(^{132}Xe/^{40}Ar)$ in ambient air, respectively.

The improved precision enabled by our inlet system should be sufficient to resolve the previously unobserved annual cycle of Kr and Xe caused by the seasonal release and uptake of both gases by the ocean as it warms and cools. The seasonal cycle of $\delta(^{40}Ar/^{28}N_2)$ has an amplitude of 5–15 per meg in the extratropics (Keeling et al., 2004). $\delta(^{84}Kr/^{28}N_2)$ and $\delta(^{132}Xe/^{28}N_2)$ however are ~3.4 and ~8.9 times more sensitive than $\delta(^{40}Ar/^{28}N_2)$ to changes in ocean temperature owing to the different temperature-dependences of Ar, Kr and Xe solubility in seawater (Hamme and Emerson, 2004; Jenkins et al., 2019). This implies that
seasonal variations in $\delta(^{84}Kr/^{28}N_2)$ and $\delta(^{132}Xe/^{28}N_2)$ have a magnitude of 17–51 and 45–134 per meg respectively, which would be readily resolved if precision of our system scales as expected with signal strength.

    The gettering inlet and MS system was applied here only for single ion (He$^+$) detection, but alternately could be applied for multi-ion collection. The acquisition of Kr and Xe isotope ratios for example would provide valuable additional information



for detecting artefactual fractionation during sampling and allow further improvements in precision by increasing the total

number of ions collected.

The need for only a single ion detector also allows the gettering and flow-stabilizing inlet to be interfaced with simpler and more affordable mass spectrometers, such as quadrupole systems. The performance of the system will depend on the stability of the $^4$He$^+$-ion beam over the time scale of switching and will need to be evaluated critically, but any variability on time scales longer than the switching time is canceled by sample-standard differencing.

## 5 Conclusions

Here, we presented a new method for high-precision measurements of changes in the He/N$_2$ ratio of atmospheric air. The method relies on monitoring of the $^4$He$^+$ ion beam in a mass spectrometer during sample-standard switching. The ion beam is stabilized by flowing sample and standard air through a single capillary into the MS from an actively pressure-controlled open-split (Henneberg et al., 1975), such that variability of the $^4$He$^+$ ion beam directly reflects differences in the helium mole fraction

of the gas mixtures. Measurements of the helium mole fraction can easily be converted to δ(He/N$_2$) after the analysis if O$_2$/N$_2$, Ar/N$_2$, and CO$_2$ concentrations of the sample are determined as well. An online getter preconcentrates He and other noble gases before entry into the MS by chemically removing >99.99 % of all N$_2$ and O$_2$ in a reaction with titanium sponge. Our method thereby avoids the need for peak jumping and avoids the need for a multi-collector mass spectrometer, while achieving a precision of ~10 per meg (1σ) on repeat analysis of δ(He/N$_2$) in high pressure tanks.

In future work, the gettering and flow-stabilizing inlet system could be used to investigate possible interannual to decadal changes in stratospheric δ(He/N$_2$) linked to variability in stratospheric circulation and stratosphere-troposphere exchange processes. Additional applications could include the search for a signal of anthropogenic helium release during fossil fuel extraction and burning (Boucher et al., 2018c; Lupton and Evans, 2013, 2004; Mabry et al., 2015; Oliver et al., 1984; Pierson-Wickmann et al., 2001; Sano et al., 1989), or measurements of spatial gradients resulting from localized human or natural

sources of helium (Boucher et al., 2018a, 2018b; Sano et al., 2010). The setup is also suitable for the analysis of other noble gases and could therefore be used to study seasonal ocean warming associated with degassing or uptake of Kr and Xe from the ocean (Baggenstos et al., 2019; Bereiter et al., 2018).

## 6 Acknowledgements

We thank Ross Beaudette, Alan Seltzer, Sarah Shackleton, Jacob Morgen, Jessica Ng, and Eric Morgan for laboratory support

and insightful discussions during the development of the He/N$_2$ analysis system. We are grateful to Shane Clark, Savannah Hatley, Adam Cox, and Timothy Lueker for providing high-pressure cylinders used during testing. We also thank them for maintaining and operating the Ar/N$_2$, O$_2$/N$_2$ and CO$_2$ analysis systems in the Keeling laboratory. This work was supported by the National Science Foundation grants MRI-1920369 and AGS-1940361.



## 7 Data availability

Data presented in this manuscript are available as an electronic supplement to this paper from the journal website.

## 8 Competing interests

The authors declare that they have no conflict of interest.

## 9 Author contribution

BB designed and build the inlet system with important design expertise from WP, JS and RK. BB performed all testing and
prepared the manuscript with contributions from all co-authors.

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



## 11 Figures and Figure Captions

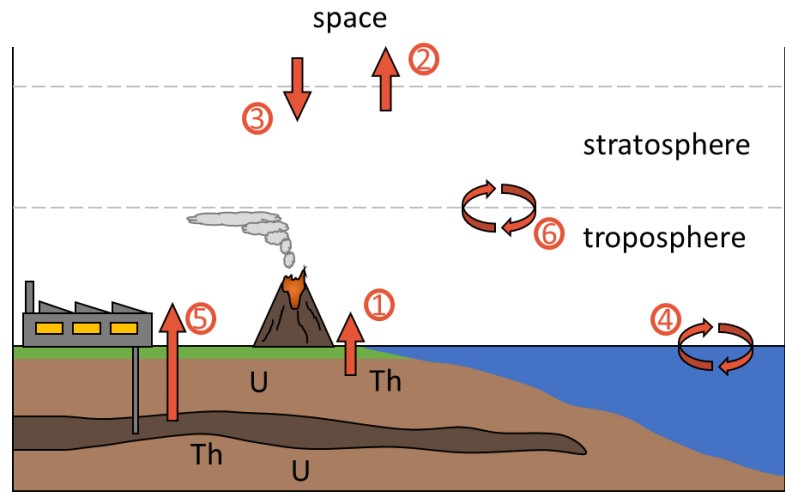

**Figure 1.** Schematic depiction of $^4$He fluxes to and from the troposphere. Different processes are numbered and listed in Table
445    1.



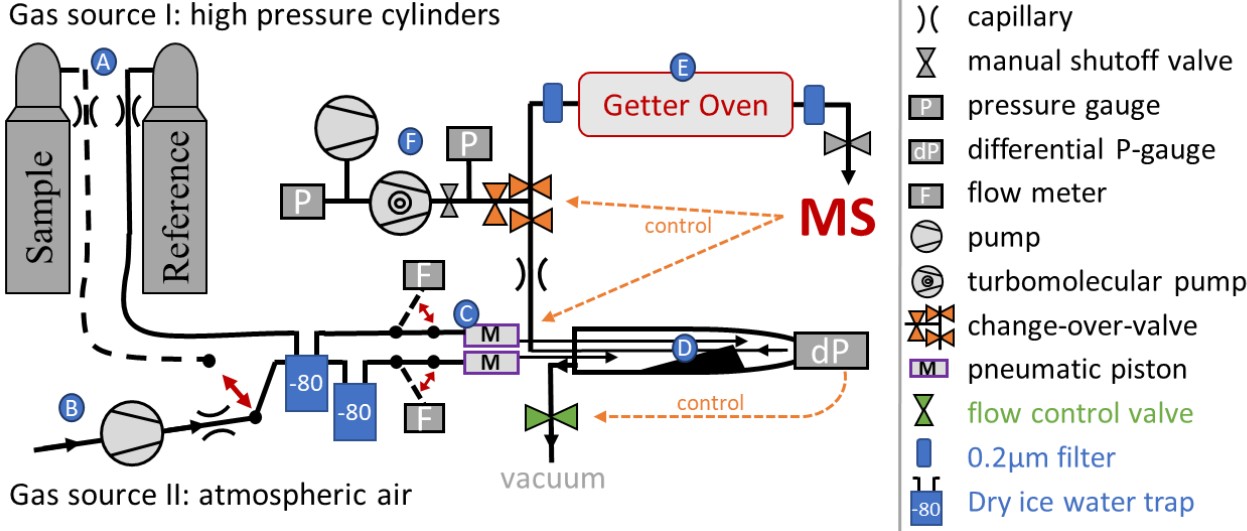

**Figure 2.** Schematic depiction of the flow-stabilizing MS inlet system. Dashed orange arrows highlighted important control pathways and letters A-F in blue circles label the main sections of the inlet system. Red double arrows indicate manual switching option in the inlet system. Gas can either be delivered from samples in high-pressure cylinders (A) or locally pumped ambient air (B). The flow can be measured by two Omron flow meters before entry into the pressure stabilization chamber (C). Pistons (D) alternatingly move fine metal tubing in the pressure stabilization chamber pushing either the sample or standard gas stream deeper into the stabilization chamber where the gas will be picked up by a single capillary leading to the MS. The chamber is exhausted to a vacuum system and the pressure is monitored and controlled by a differential pressure gauge combined with an automatic MKS flow control valve. The stainless-steel getter oven (E) has an inner diameter of 1/2" and is filled with 10–12 g of titanium sponge. 2μm-filters prevent particles from contaminating the MS and gas delivery system. In case of an anomalous pressure change in the MS or when venting the getter oven, the getter oven can be isolated from the pressure-stabilization chamber with a change-over-valve controlled directly by the MS software. The entire inlet vacuum system is backed by a diaphragm vacuum pump and a turbomolecular pump (F). A manual shutoff-valve can isolate the getter oven from the MS.





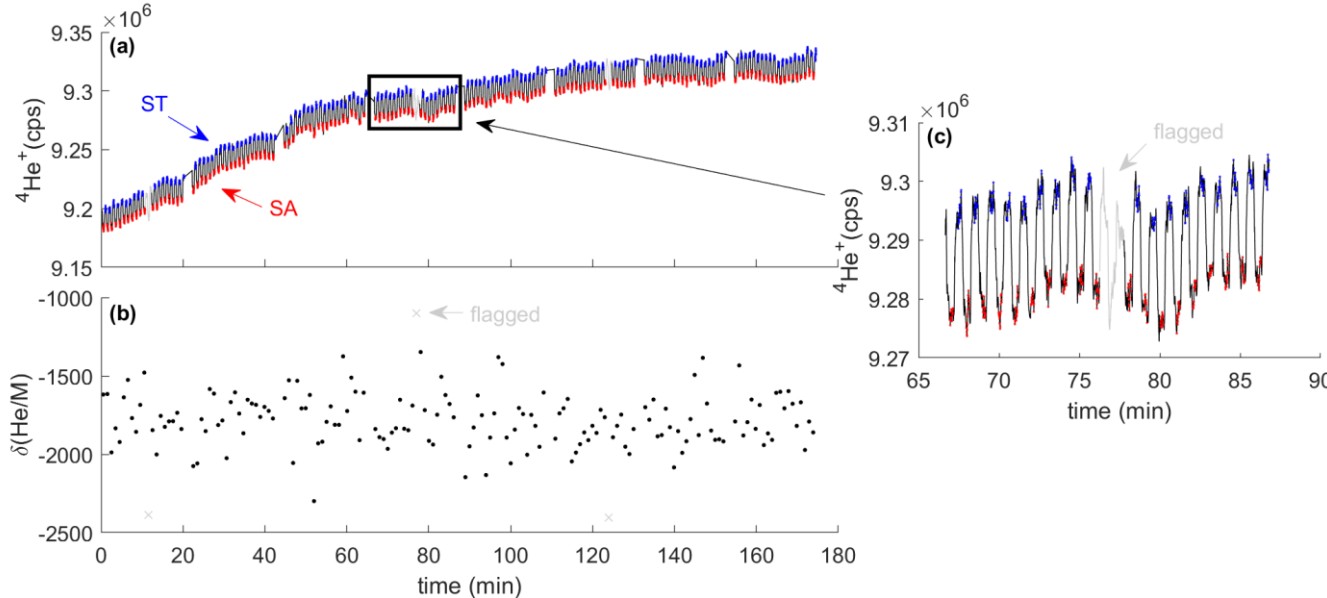


**Figure 3.** Typical analysis results from the measurement of two high pressure cylinders. The MS monitors the $^4$He$^+$-ion beam during switching between sample (SA) and standard (ST) gas (a). Red and blue shaded data points highlight the periods used for integration and calculation of the delta value (b). They are separated by idle times (black lines) to allow complete flushout after switching. Data are quality controlled and flagged periods are shown in grey. Inset (c) shows one block of 20 sample-

standard comparisons including one cycled that was flagged as an outlier.

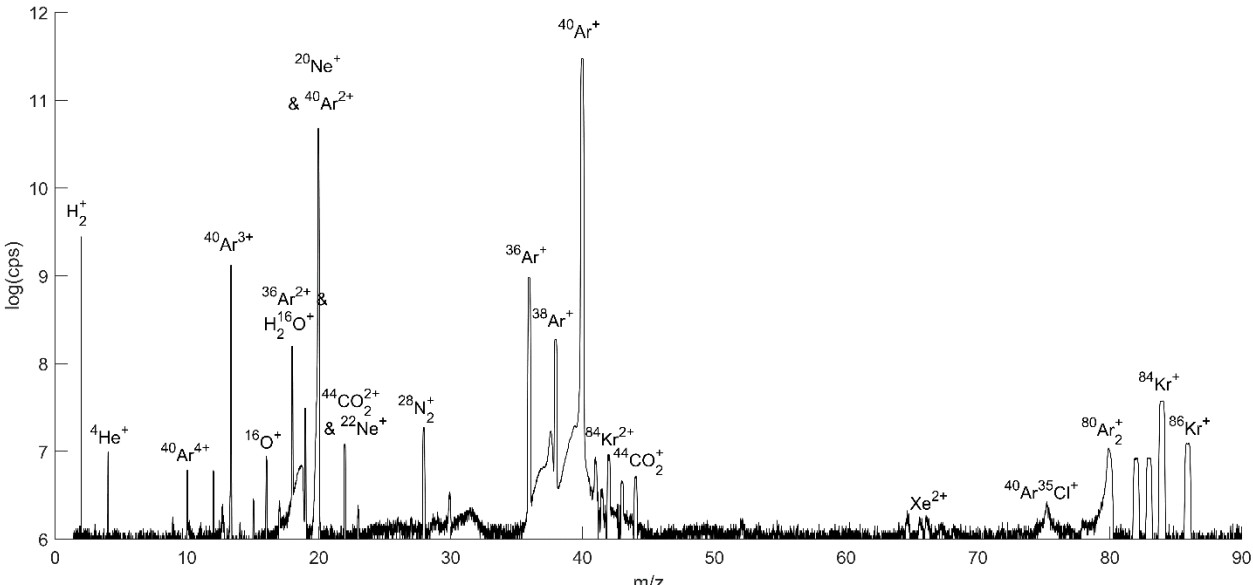

**Figure 4.** Mass scan of ambient air. Ion beam intensity is shown as the logarithm of the ions counted per second, and select ion species are labeled.

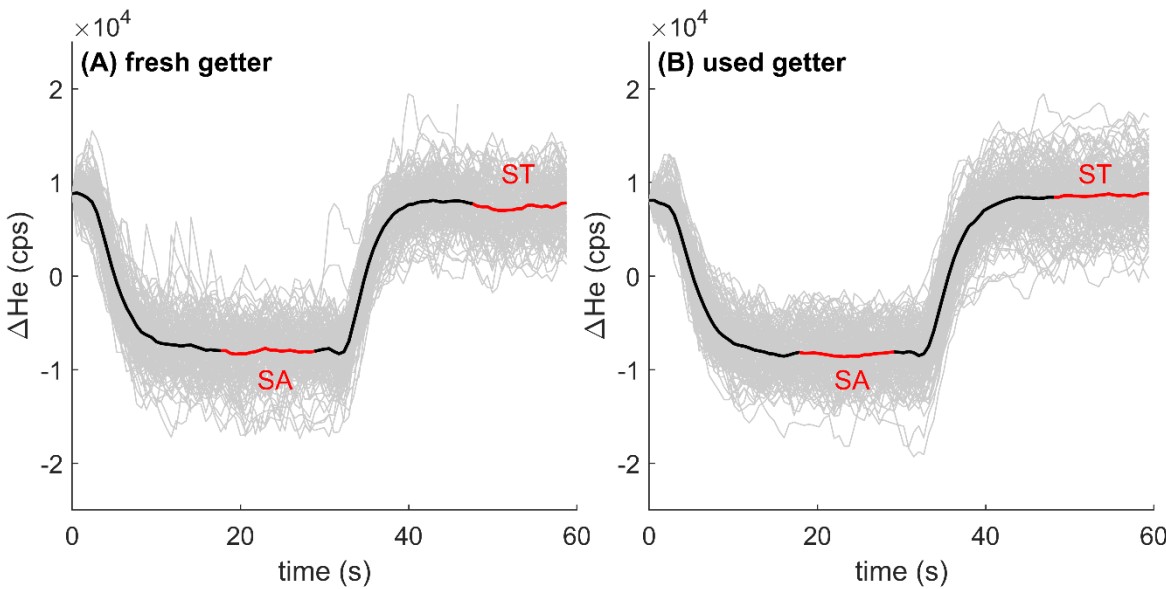

**Figure 5.** Stack of $^4$He ion count difference ($10^4$ counts per second, cps) when switching between the same standard (ST) and sample (SA) gas stream using fresh titanium sponge (A) and nearly depleted getter material (B). Grey lines show individual records forced to align at time equals zero and the thick black line shows the average of all stacked switching events. The analysis cycle consists of (i) switching to SA with an idle time of ~18 seconds, (ii) a ~12 second integration of ions from SA, (iii) switching back to ST, again with a ~18 second idle time, and finally (iiii) a ~12 second integration of ST.





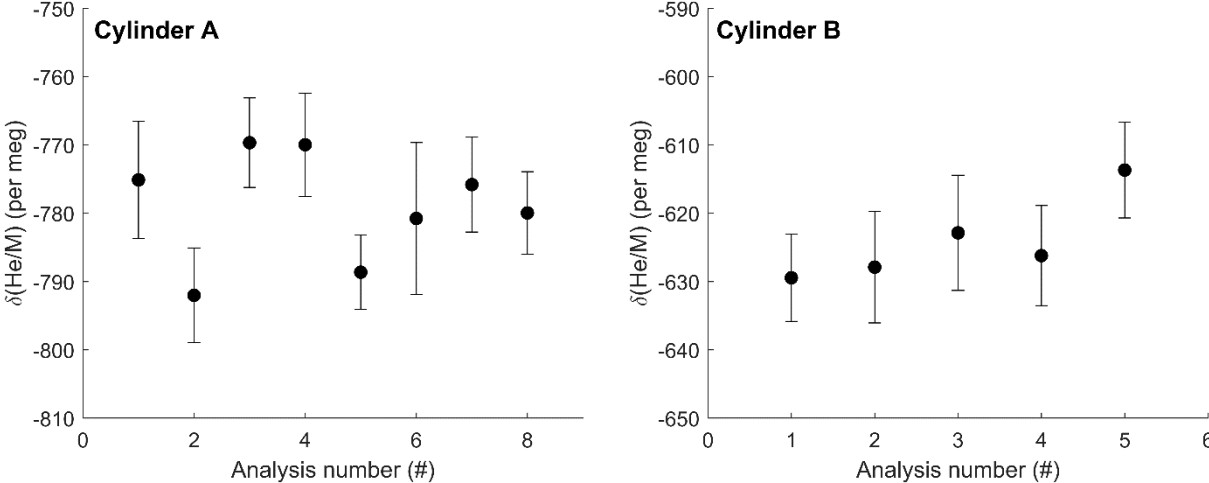


**Figure 6.** Repeat δ(He/M) measurement of two high-pressure cylinders against ambient La Jolla air collected in 2019. Repeat analysis show a standard deviation of 8.1 and 6.3 per meg for cylinder A and cylinder B respectively. Analysis 6 for cylinder A was shorter resulting in a larger uncertainty for that measurement. Data are not corrected for zero enrichment effects discussed in the text.



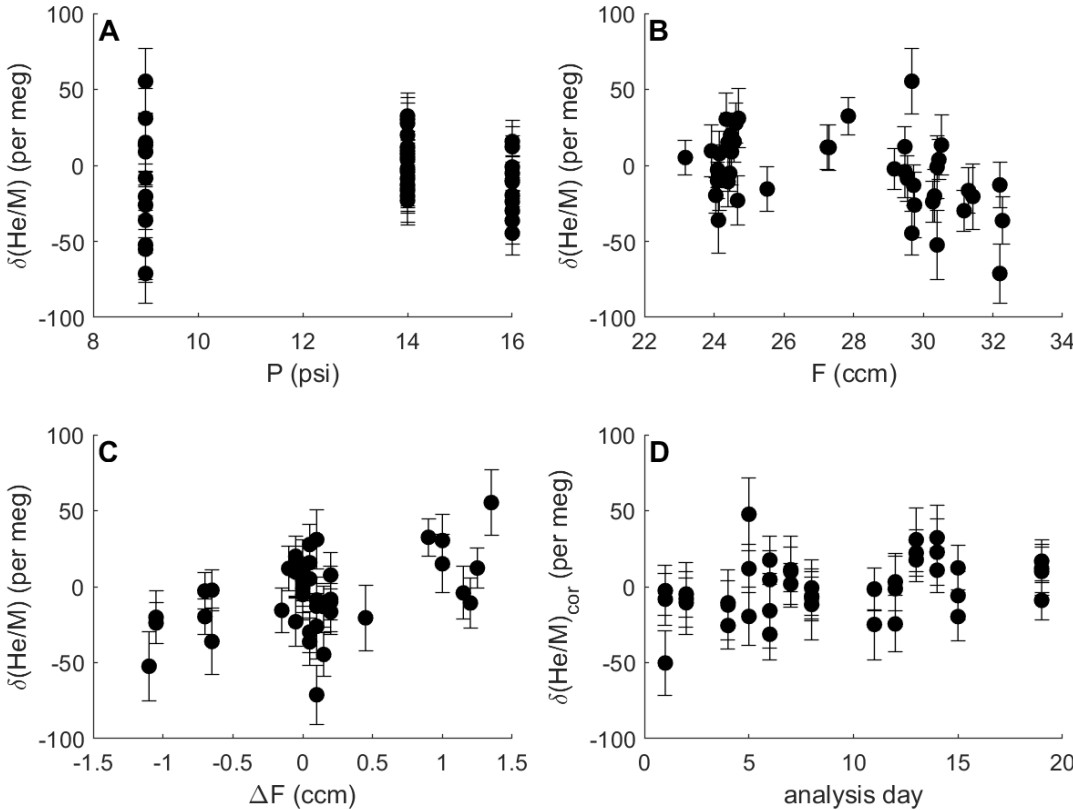


**Figure 7.** Difference in δ(He/M) between two identical gas streams (i.e., the zero enrichment) measured repeatedly under different conditions over 1.5–3h. Error bars show 1σ uncertainty. Measurements were made at different pressure levels (a), with slightly varying gas flows to the stabilization chamber (b), and imbalances in flow between SA and ST side (c). The same shared capillary was used for all analysis. Therefore, the pressure in the stabilization chamber controls the intensity of the ion

beam and the internal precision of the analysis, illustrated by the greater scatter of observations at 9 psi (62.1 kPa). Delta values shown in (d) were corrected for the influence of pressure, mean flow, and flow imbalance according to coefficients found by multiple linear regression (see text). For a pressure of 14 psi (96.5 kPa), corrected delta values generally show scatter as expected from shot-noise behavior and corrected delta values are stable over time.



## 12 Tables and Table Captions

**Table 1.** Processes contributing to variations in the tropospheric and stratospheric $^4$He/N$_2$ ratio.

| Process | $^4$He flux (10$^7$ mol y$^{-1}$) | Tropospheric δ($^4$He/N$_2$) trend[1] (per meg y$^{-1}$) | Stratospheric δ($^4$He/N$_2$) trend[1] (per meg y$^{-1}$) | Tropospheric δ($^4$He/N$_2$) anomaly (per meg) | Reference[2] |
|---|---|---|---|---|---|
| *Long-term trend* | | | | | |
| (1) Crustal degassing and volcanism | 24.0–50.7 | 0.26–0.55 | | | (a) |
| (2) Loss to space | 53.3–106.8 | -0.58–-1.15 | | | (a,b) |
| (3) Non-terrestrial sources | insignificant | - | | | (a) |
| (4) Global Ocean warming[3] | 1.3 | -0.16 | | | |
| (5) Fossil fuel extraction[4] | 3189–12755 | 34–138 | | | (c) |
| | 13000±7000 | 140±76 | | | (d) |
| | 34000 | 367 | | | (e) |
| (6) BDC acceleration[5] | | 0.5 | -15 | | |
| *Observational limits on decadal trends[6]* | | -1.4±44.5 | | | (f) |
| | | 9.5±32.7 | | | (g) |
| | | -2±23.8 | | | (h) |
| *Seasonal and interannual variability* | | | | | |
| Seasonal cycle of global ocean heat[7] | | | | 3–9 | |
| Strat. circ. & STE variability[8] | | ±6 | ±375 | | |
| *Interhemispheric difference[9]* | | | | <30 | |

[1] δ($^4$He/N$_2$) trends are calculated using first column and assuming total atmospheric $^4$He = 9.268 e+14 mol. N$_2$ changes are generally neglected except for ocean degassing. Tropospheric trends are globally uniform because the troposphere is well mixed. Stratospheric trend estimates are given for 35km in the mid latitude Northern Hemisphere.

[2] (a) Torgersen (1989)    (d) Pierson-Wickmann et al. (2001)    (g) Mabry et al. (2015)
(b) Kockarts (1973)    (e) Sano et al. (2013)    (h) Boucher et al. (2018c)
(c) Oliver et al. (1984)    (f) Lupton and Evans (2013)

[3] calculated from $^4$He and N$_2$ solubility changes (Weiss, 1971; Hamme and Emerson, 2004) for an ocean heat content trend of 10ZJ y$^{-1}$ at a mean water temperature of 10°C.

[4] (c) includes natural gas, coal and uranium, (d) and (e) include natural gas, petroleum and coal.

[5] δ($^4$He/N$_2$) rescaled from δ(Ar/N$_2$) assuming 7.5x greater gravitational separation. The secular δ(Ar/N$_2$) trend was simulated in the SOCRATES model for an accelerating BDC scenario (+4 % dec$^{-1}$) by Ishidoya et al. (2020). δ($^4$He/N$_2$) trend is adjusted to reflect a more plausible BDC acceleration of +2 % dec$^{-1}$.

[6] observed $^3$He/$^4$He trends are translated to $^4$He trends assuming $^3$He/$^4$He = 3e-8 in fossil fuel associated helium.

[7] scaled from seasonal δ(Ar/N$_2$) changes of 5-15 per meg (Keeling et al., 2004) using solubility-temperature dependency of He, N$_2$ and Ar in a 10°C warm surface ocean (Weiss, 1971; Hamme and Emerson, 2004).

[8] Tropospheric and stratospheric δ($^4$He/N$_2$) rescaled from δ(Ar/N$_2$). Ishidoya et al. (2020) report a ±0.4 and ±25 per meg δ(Ar/N$_2$) change in troposphere and stratosphere in the SOCRATES model for a sinusoidal ±5% change in BDC strength over 3 years.





[9] Assuming that industrial He release is confined to the Northern Hemisphere and assuming an annual $\delta(^4\text{He}/\text{N}_2)$ increase of $\sim$30 per meg (consistent with the current observational error) yields an interhemispheric $\delta(^4\text{He}/\text{N}_2)$ difference < 30 per meg. Differences in STE of He between the hemispheres are neglected here but could be important.





**Table 2**. Summary of observed ion beams in Figure 4. Relative ion beam intensities on MAT253 are calculated from the scan with identical source tuning. Xe isotope beams were not observed but scaled from previous observations in the lab.

| m/z | Dominant ions | Ion beam intensity (cps)[*] | ion beam intensity relative to He[+] |
|---|---|---|---|
| 4 | $^4He^+$ | 9.70E+06 | 1 |
| 20 | $^{20}Ne^+, ^{40}Ar^{2+}$ | 4.78E+10 | 4916.1 |
| 22 | $^{22}Ne^+, ^{44}CO_2^{2+}$ | 1.22E+07 | 1.25 |
| 36 | $^{36}Ar^+$ | 9.55E+08 | 98.26 |
| 38 | $^{38}Ar^+$ | 1.89E+08 | 19.44 |
| 40 | $^{40}Ar^+$ | 2.98E+11 | 30660 |
| 82 | $^{82}Kr^+$ | 8.50E+06 | 0.87 |
| 83 | $^{83}Kr^+$ | 8.40E+06 | 0.86 |
| 84 | $^{84}Kr^+$ | 3.76E+07 | 3.87 |
| 86 | $^{86}Kr^+$ | 1.22E+07 | 1.25 |
| 129[*] | $^{129}Xe^+$ | 2.60E+06 | 0.27 |
| 131[*] | $^{131}Xe^+$ | 2.10E+06 | 0.22 |
| 132[*] | $^{132}Xe^+$ | 2.70E+06 | 0.28 |
| 136[*] | $^{136}Xe^+$ | 9.00E+05 | 0.09 |

[*]Xe isotopes were not measured directly here because of the limited dynamic range of the MAT 253 when set to measure He. Instead we
report expected Xe ion beam intensities scaled from previously analysis of Kr and Xe in the lab assuming natural isotopic abundances.