# Peer review of "A method for resolving changes in atmospheric He/N2 as an indicator of fossil fuel extraction and stratospheric circulation"

_Atmospheric Measurement Techniques, 2020_

## Referee Comment (RC1) · Anonymous Referee #1 · 9 Oct 2020

This is a very well-written manuscript describing a new technique for an improved quantification of He/N2 ratios in air samples; a development that could contribute to answering many important questions in tropospheric and stratospheric science. My main criticism is that there is a lack of detail in places, but I would recommend publication after the comments below have been dealt with appropriately.

L24 Why presumably? Are there other possible causes e.g. discussed in the 7 papers cited here?

L34 14C is neither a trace nor a greenhouse gas.

L42-44 This is correct for the tracers listed here, but not for more recently introduced

ones in e.g. Leedham Elvidge et al., ACP, 2018. Also, as stratospheric air can be several years old, these gases are governed by their respective tropospheric growth rates and its variability over time (and so is the He/N2 ratio).

L50 Isn't this limit dependent on the amount of air used?

L53 This should probably be just Boucher et al., 2018c.

L82 Please cite Fuller et al. properly.

L96 Does "the level of roughly ±30 per meg per year" describe the analytical precisions or the actual trend? Also, consider splitting this very long sentence into two.

L110 This should be "on the order of" or perhaps "below" (considering Table 1, where estimates in the 10^8 range are listed). Do these estimates include the impact of large volcanic eruptions?

L124 "Fig. 2C" creates the impression that the figure consists of several separate sub-figures which is not the case. Consider rephrasing e.g. to "C in Fig. 2".

L124-126 It's not clear to me from this description a) how the pistons move the tubing, b) how stress to the tubing moving outside the chamber is dealt with and c) how often this part of the system develops leaks (or whether this has been checked at all). See also my comment on the caption of Fig. 2.

L130 Please specify which vacuum grease is used.

L131 Why does the outlet capillary have to be thermally insulated? What is the required temperature stability? Also, to which diameter has the capillary been crimped and why?

L134 Please quantify "high purity".

L145-146 Which criteria are used to judge whether the Ti needs replacing? What are the volume and dimensions of the getter oven and how is it heated?

L152-154 How is a "complete flush-out" ensured? What is the internal volume of the

flushed-out parts? Shouldn't the getter oven be flushed for longer since the flow in there breaks down by a factor of ~100?

L171 How does one adjust the crimping of the capillaries, especially without risking to break them?

L165-166 At which temperature?

L166-167 That sounds dangerous. How is compliance with Health and Safety regulations ensured?

L171-173 What is the natur e of this cold trap and how is its temperature stability ensured? What is its internal volume and how often does it need cleaning/exchanging? What mechanisms are in place to prevent ice building up in the vicinity?

L186 Have any other cycle times been tested?

L203-205 If the uncertainty of the correction is 6 per meg, why does it increase the analytical uncertaintyby only 2 per meg? Also, what about the uncertainty of 1.5 h measurements?

L211-213 What are the uncertainties of the corrections?

L216 I'm not sure that the derivation of this relationship should be part of a Discussion section as it's rather a methodological part of the manuscript. Also, how much larger than relative changes in $N_2$ are relative changes in $CO_2$?

L217-265 This section focuses on the multitude of possibilities that the new technique might enable, but fails to draw much attention to its limitations (e.g. the large amount of air required, which is not easily acquired from the stratosphere, or the long analysis times, which will limit constraints on spatial gradients in volcanic plumes or near oil or gas facilities).

L235 Do the authors perhaps mean "possible complications that affect $3He$ measurements" here?

L273 2 * "avoids the need"

L278-282 I'm not sure that it's necessary to list all these references again. It certainly doesn't help the readability.

L451-453 Very little detail is provided (in both the figure and the caption) on how the piston system looks and operates. Since this is a crucial part of the inlet system I urge the authors to expand their explanation and perhaps include a cross-section of this chamber. Also, how does the pneumatic control work?

L477 How many sigmas?

L490 Table 1 has a lot of empty space and at the same time an enormous amount of footnotes, which make it rather difficult to follow. Consider reorganising, e.g. through naming the references in the table or moving some of the explanations into the caption. Also, what uncertainty range is connected to the assumption of 3He/4He = 3e-8?

---

## Referee Comment (RC2) · Anonymous Referee #2 · 13 Oct 2020

Review AMT Manuscript "Measurements of atmospheric He/N2 as an indicator of fossil fuel extraction and stratospheric circulation"

General assessment

The manuscript presents a method for analysis of the He mole fraction in large air samples, calibrated relative to a gas standard. The method allows much better precision than previous methods for He analysis in air. As far as I can tell from the manuscript, the analytical tests are reliable and demonstrate the instrumental performance of the method in a good way. However, in the current form of the manuscript, the scientific relevance of the new method remains unclear to me, and I have some technical questions, which deserve more background explanation. If these points can be fixed in a revised version, I recommend publication of the manuscript in AMT.

Specific comments

* The measurement yields the mole fraction of He in the sample gas matrix, normalized relative to the He mole fraction in a gas standard. However, the method is presented in the manuscript to yield the (absolute?) He/N2 ratio of the sample. This is rather confusing, as N2 is NOT part of the measurementn (in fact, N2 is removed from the gas sample before MS analysis). The conversion of the measurement result (He mole fraction) to the He/N2 ratio is based on a simple mathematical manipulation and some assumptions about the composition of the gas matrix. This conversion seems trivial and is not related to the measurement technique; and I don't see the need for it. I suggest to avoid this confusion by removing He/N2 (or He/O2 etc.) as far as possible (from the title and most of the text), and to focus on the true nature of the measurement, i.e., on the (relative) He mole fraction.

* The manuscript is missing a review/overview of existing techniques for He analysis in air (3He/4He, He/Ne, He mixing ratio). Without this background, it is difficult to understand on what grounds the new method was designed, and how it improves on previous methods, both in terms of instrumental techniques and scientific applications. The manuscript should be revised to better develop the link of the new method to existing techniques. What were the design targets for the new instrument? What was the design approach to achieve these targets? Why was the new method implemented in this way?

* As it is, I am not convinced about the scientific utility of the new method: (1) Earlier work with static MS systems for He isotope analysis showed that their 0.2% precision was sufficient to resolve the expected atmospheric He variations (for example: Mabry et al, EPSL, 2015). Why exactly is a new method with approx. 100x better precision required to study the evolution of the atmospheric He? (2) Most publications on atmospheric He acknowledged that the quality of the historic/archived air samples put major limitations on the uncertainty of reconstructing the atmospheric He abundance (for an overview see for example: Brennwald et al, EPSL, 2013 / reference missing in the manuscript). Further improving the instrumental precision of the He analysis does not help with this fundamental issue. I am therefore not convinced that an improvement of the instrumental precision is very useful to study some of the effects noted by the authors. (3) With an analysis time of 6-8 hours and a sample gas consumption of 28 mL per minute, a single analysis requires about 10-14 L of sample gas. However, typical samples of historic/archived air are typically in the order of 0.1-100 mL (see refs. cited in the manuscript, and as given above), which is 2 to 5 orders of magnitude too low for the new method described in the manuscript. The large gas consumption of the new method therefore severely limits the technical applicability to large-volume gas archives, which are scarce and may not be readily available for consumption to a (destructive!) analysis. In order to illustrate the scientific potential of the new method, the manuscript should be extended with application examples targeted at the scientific questions described in the introduction. I suggest to add a few measurements of real-world historic/archived air samples to demonstrate the true utility and suitability of the new method to study the He change due to fossil fuel extraction (and analogous for stratospheric circulation).

* What is the purpose of the cold traps at the gas inlet system? My guess is that they are meant to remove water vapor from the sample/standard gas streams, but I am not sure. This should be described better.

* The intensity of the He ion beam in the MS is controlled by the sample/standard gas flow to the gas inlet system. As far as I can tell, the precision of the analysis result is therefore controlled crucially by the inlet system, and in particular by the cold traps and the pressure regulators: (1) Cold traps: the efficiency of the water removal from the gas streams is likely not stable over time, and will therefore introduce a variation of the He mixing ratio in the gases. How can the cold trap variation be avoided such that

the He mixing ratio in the gas stream is stable to 10 per meg or better? (2) The gas pressure at the main capillary inlet directly controls the gas flow into the capillary, and hence the He flow into the MS. Similarly to the cold traps, this seems to imply that the pressure at the capillary inlet must be stabilized to 10 per meg or better. How is this technically possible? I don't have a strong background with large-volume dynamic MS gas analysis, so I may be overlooking something that may be obvious to the authors (I am mostly into static and low-volume dynamic MS). However, I believe it would be very difficult to achieve such tight stability controls over the gas inlet system, and there must be some way to avoid or compensate such variations in the gas inlet in order to achieve the 10 per meg precision in the He mixing ratio of the gas sample. I feel these points need to be explained better.

* Title: "Measurements of ..." seems to indicate that the focus of the paper is to present new measurement data. I'd suggest to change focus to the new analysis method / technique.

* Line 26/27: "...gases heavier than air in the stratosphere..." –> what gases other than air are in the stratosphere? Also, does the gravitational separation only apply to heavier gases, not lighter ones?

* Line 56/57: I'd assume that N2 and other "noble" gases are seasonally variable due to atmosphere/ocean gas exchange, which is subject to the temperature dependence of N2 solubility in the water (e.g., Keeling et al., Tellus 322–338, 2004). Note that the He solubility dependence on temperature is much less than for N2. While this effect may be relatively small, it should at least be noted in the text (if the N2 normalization is not removed entirely, see above).

* Line 83: Using the equations given above, I calculate del(He/N2) = -6.4, not 7.5 (note the sign!). Please check.

* Line 95: "...no significant trend in atmospheric 3He/4He has been observed..." –> there are many different studies with different (and sometimes contradicting!) conclusions regarding the existence and size of the atmospheric 3He/4He change. Considering the work cited in the manuscript it seems wrong to say "no significant trend was observed", and this argument needs to be revised. To this end, it might be useful to take a look at Brennwald et al, EPSL, 2013, which has a compilation and comparison of different studies on the atmospheric 3He/4He change, and also presents some measurements on He changes observed in archived air samples.

* Line 144: "...compared to the 30-second switching timescale." –> what does this refer to? Switching of what?

* Line 156: "Background concentrations..." –> what "background" is this? "Blank"? "Residual"? m/z=4 signal with analysis of He free gas? Or with the inlet to the MS closed? Please define.

* Line 172: The geometry of the cold trap crucially controls the operation and performance of the cold trap. Is this a U shaped tube? Or a 'washing flask' type? Or something else? Please explain the details of the cold traps ("made from 1/4" stainless steel" is not sufficient).

* Line 175–178: Why is the performance of the getter important (as long as it is not "dead" and works as a pump to draw the air matrix into the capillary)? As far as I can tell, the He flow rate into the MS is identical to the He flow rate into the getter. I therefore don't see how the getter can affect the He analysis (as long as the getter operation is stable between the analysis runs of the sample and standard gases). How does the getter affect the performance of the He analysis? Does the getter performance affect the He analysis at all? Please explain.

* Line 201: What does "the 5% level" refer to? Is this a statistical "significance level"? What kind of statistics? Please explain.

* Fig. 7 shows the measured zero effect, with a substantial scatter (about +/- 50 per meg or so). However, the discussion in the main text indicates a zero-effect uncertainty

of only 6 per meg. I am confused because I don't understand how the +/- 50 per meg scatter is consistent with the 6 per meg uncertainty. I believe my confusion is due to some ambiguity in how the zero-effect is quantified and compensated during the analysis and data processing routines. This should be explained better.

* Tab. 2: what is the meaning of "scaled" Xe peak heights? What kind of "scaling" was applied, and how? Please explain.

---

## Referee Comment (RC3) · Anonymous Referee #3 · 20 Oct 2020

In this paper, the authors describe a new measurement system they developed to precisely measure atmospheric He/N2. It is my opinion that the unprecedented high precision achieved by the method will make a significant contribution to an increased understanding of the He budget and gravitational separation processes in the stratosphere. The authors' effort is to be congratulated. I have found the paper to be well written and should be published in Atmos. Meas. Tech.. However, listed below are some comments I would like the authors to address before the publication.

1) I think the analytical precisions of mass spectrometry can be classified into "internal precision", "internal reproducibility" and "external reproducibility" (see Bender et al.,

1994, Geochim. Cosmochim. Acta.). Internal precision indicates a standard error of xx cycles in a 1-block analysis, and internal reproducibility indicates a standard deviation of the repeated analyses of several blocks. I think the authors did not distinguish between internal precision and internal reproducibility in the paper, and the presented internal precision of ±15 per meg for sample run of 1.5h and ±8 per meg over 6h (line 186-187) are standard error (1sigma/sqrt(n), n = number of the total cycles ∼ about 1,800 and 7,200 for 1.5h and 6h, respectively). The internal precision is shown as error bars in Fig. 6, and the variability of every cycle measurements is presented in Fig. 3 (b) (the unit "per meg" is needed for the y-axis label of Fig. 3 (b)). I hope I have not misunderstood or misinterpreted the presentation.

2) Fig. 6 shows d(He/M) values of 8 (5) times repeated analyses of the 6-8h measurements of Cylinder A (B) against La Jolla air. I think it corresponds to the external reproducibility in the context of Bender et al., but I have some concerns: Is the La Jolla air identical to the "He/N2 reference material (line 163-165)", "standard gas cylinder (line 188)", and "reference (Fig. 2)"? Please clarify to avoid confusion. Assuming that the standard and La Jolla air are identical and used as the reference in Fig. 2, how do the authors ensure the short-to-long term stability of the La Jolla air? In other words, how long does it take to obtain the results of 8 or 5 times repeated measurements shown in Fig. 6? If it takes several days, then I agree that the stability of the standard is enough for the period, however, future work is needed to confirm much longer-term stability for the observations of seasonal and interannual variations in the atmospheric He/N2 ratio.

3) Please provide information related to the needed minimum and maximum inner pressures of the sample and reference gas to maintain the appropriate flow rate of 27-28 mL/min using capillary. In this regard, I sort of remember Scripps flask samples being collected at an atmospheric pressure. If so, I don't think these flasks can be used for the He/N2 analysis, as described in the present study. How do the authors collect pressurized air samples without significant fractionation of He and N2?

4) Related to the comment 3) above, there does not seem to be any discussion of the analyses of the locally pumped ambient air shown in Fig. 2. I would be very much interested in hearing from the authors some information regarding the ambient air analyses, since it will serve as an indicative evaluation of "true" external reproducibility, including air sampling procedures, such as inlet fractionation, and leakage or permeation of He through pump diaphragm.

5) I think the response of d(He/N2) measured by the mass spectrometer to the actual H2/N2 ratio of a sample air will likely be linear, but some checks are needed. Please provide some concrete evidence to that effect.

6) The potential application of He/N2 to evaluate interannual variability in the stratospheric circulation ($\pm375$ per meg/yr, expected value) is very interesting and is an excellent idea, considering that it can provide better signal-to-noise ratio than Ar/N2. In order to extract the gravitational fractionation signal of He/N2, it is necessary to subtract the He/N2 change due to chemical processes such as fossil fuel extraction (which yields 35 – 350 per meg/yr of secular He/N2 trend at the surface) from the observational results in the stratosphere. I guess the authors have plans to observe surface He/N2 variations, so that a precise secular trend in the troposphere will be achieved in the near future. However, as reported by Engel et el. (2009 Nature Geoscience), CO2 or SF6 age of the stratospheric air samples show interannual variability by about $\pm1$ years. This interannual variability of the age corresponds to $\pm35 - \pm350$ per meg of the stratospheric He/N2 change due to the fossil fuel extraction, if we ignore attenuation of the interannual variability from the surface to the stratosphere. Therefore, to evaluate the interannual variability in the stratospheric circulation based on gravitational fractionation of He/N2, precise determination of He/N2 age will be important. However, as Ray et al. (2017 JGR) and Sugawara et al. (2018 ACP) reported, CO2 age and SF6 age do not necessarily agree with each other. Given these issues, I would be very much interested in hearing how the authors are planning to determine the age of He/N2.

7) Line 198: "9 psi, 14 psi, and 16 psi. . ." should be changed to "9, 14, and 16 psi".

8) Line 202: "-9.61±7.2 per meg, 1±3.7 per meg, and -15.7 per meg..." should be changed to "-9.61±7.2, 1±3.7, and -15.7 per meg...".

9) References: Please change "2" in CO2, O2/N2, Ar/N2 and SF6 to subscripts.

---

## Author Comment (AC1) · 6 Jan 2021

Please see attached the authors' response to reviewers including a document tracking changes to the manuscript.

Please also note the supplement to this comment:
https://amt.copernicus.org/preprints/amt-2020-313/amt-2020-313-AC1-supplement.pdf

---

## Author Response (AR1)

**Response to reviewers**

We thank the three anonymous reviewers for their excellent and insightful feedback on the manuscript. The most significant changes made to the manuscript at this stage include providing more detail in the descriptions of the piston and pressure stabilization system as well as the water traps and adding a section that discusses the current sample requirements. In the following we address specific comments individually.

**Anonymous Referee #1**

L24 Why presumably? Are there other possible causes e.g. discussed in the 7 papers cited here?

We thank the reviewer for pointing out this ambiguity. We rephrased the sentence to focus on the lack of direct observational evidence instead. Helium isotope studies have placed some constraint on an atmospheric $^3$He/$^4$He trend but are limited by analytical uncertainty and uncertainty about the stability of 3He in the atmosphere ( e.g., Boucher et al., 2018). Evidence from helium isotope studies is discussed in more detail later in the manuscript.

L34 14C is neither a trace nor a greenhouse gas.

Yes, we changed the phrasing to "societally-important greenhouse gases and geochemical tracers such as […]" in order to encompass $^{14}$C.

L42-44 This is correct for the tracers listed here, but not for more recently introduced ones in e.g. Leedham Elvidge et al., ACP, 2018. Also, as stratospheric air can be several years old, these gases are governed by their respective tropospheric growth rates and its variability over time (and so is the He/N2 ratio).

We thank the reviewer for bringing up this issue. New tracers of stratospheric age of air are needed as exemplified by the work of Leedham Elvidge et al., (2018). We hope that He/N$_2$ may become one of several new tracers that will become a useful tool to study stratospheric circulation because each tracer is characterized by a unique set of advantages and limitations. We modified the paragraph to also mention the role of the tropospheric increase of He/N$_2$.

L50 Isn't this limit dependent on the amount of air used?

In theory, the use of very large air samples could improve the precision of $^3$He/$^4$He measurements. However, in practice the use of static vacuum mass spectrometers places important limits on sample size.

L53 This should probably be just Boucher et al., 2018c.

Yes, we corrected the text.

L82 Please cite Fuller et al. properly.

We clarified that we are citing a method presented by Reid et al. (1987).

L96 Does "the level of roughly ±30 per meg per year" describe the analytical precisions or the actual trend? Also, consider splitting this very long sentence into two.

Observed trends of atmospheric $^3$He/$^4$He in recent studies are scattered near zero and have an uncertainty of about 30 per meg. We have split the sentence and clarified the language.

L110 This should be "on the order of" or perhaps "below" (considering Table 1, where estimates in the 10ˆ8 range are listed). Do these estimates include the impact of large volcanic eruptions?

We corrected the typo. Note that the estimate refers to moles of $N_2$. Due to the large abundance of $N_2$ in the atmosphere, only changes at the level of 10^14 mol y$^{-1}$ or greater would have a significant impact on $^4$He/$N_2$. Volcanic emissions, even from large volcanic eruptions, are unlikely to reach that level.

L124 "Fig. 2C" creates the impression that the figure consists of several separate sub-figures which is not the case. Consider rephrasing e.g. to "C in Fig. 2".

Thank you for pointing this out. We adopted the proposed phrasing.

L124-126 It's not clear to me from this description a) how the pistons move the tubing, b) how stress to the tubing moving outside the chamber is dealt with and c) how often this part of the system develops leaks (or whether this has been checked at all). See also my comment on the caption of Fig. 2.

We thank the reviewer for this important question and have provided more detail about the pistons and the stabilization chamber in the text. We also modified Fig. 2 to show a more detailed schematic of the sliding seal setup to address this issue. Note that the sliding tubing does not need to make a high-vacuum seal because pressure in the stabilization chamber is only varied between 6-16 psi and actively maintained. The sliding seal (as all aspects of the setup) were screened for leakage by spraying pure He around potential weak points and we have never experienced any leakage.

L130 Please specify which vacuum grease is used.

We added the information to the manuscript.

L131 Why does the outlet capillary have to be thermally insulated? What is the required temperature stability? Also, to which diameter has the capillary been crimped and why?

Changes in temperature lead to changes in conductance of the capillary. Temperature fluctuations could therefore erode the air flow stabilization. Temperature fluctuations at the crimp have not been quantified but empirically, the addition of more thermal mass to the crimped tubing has improved the stability of the $^4$He beam. The capillary diameter has also been established empirically. The degree of crimping was chosen to yield the maximum air flow that could be safely accepted by the mass spectrometer without risking frequent arcing due to excessive pressure in the MS source.

L134 Please quantify "high purity".

We added the information to the manuscript.

L145-146 Which criteria are used to judge whether the Ti needs replacing? What are the volume and dimensions of the getter oven and how is it heated?

We agree with the reviewer that this is important information and have added it to the text under section 2.2. Ti is replaced when the gettering capacity is exhausted and $N_2$ begins to break through as can be monitored from an increase in MS source pressure.

 How is a "complete flush-out" ensured? What is the internal volume of the flushed-out parts? Shouldn't the getter oven be flushed for longer since the flow in there breaks down by a factor of ~100?

Flush-out time is determined by the pressure and volume flow of gas through our inlet and determined empirically. The flush-out time is equivalent to the switching response time scale of inlet system seen in Fig. 5. After removal of $O_2$ and $N_2$ in the getter, the volume flow is roughly 100x smaller, but pressure drops simultaneously ensuring that the purging time of the getter is comparable to the tubing upstream. The empirically determined flush-out time is consistent with the volumes and flows seen in the system. We have changed the phrasing in the text to shift the focus of the reader towards the time scale of returning to a constant MS signal after switching, which is the ultimate goal. Furthermore, we clarified the importance of the pressure drop in the getter oven.

L171 How does one adjust the crimping of the capillaries, especially without risking to break them?

Stainless steel capillary tubing is squeezed between two pieces of aluminum: a flat piece and a dowel. Crimping can be adjusted by adjusting two screws which hold the pieces of aluminum together. Relaxing the force applied allows the capillary tubing to spring back. If necessary, the relaxation can be assisted manually with pliers by gently squeezing the tubing at a 90 degree angle to the direction of the crimping force.

L165-166 At which temperature?

Cylinders are stored at room temperature in the thermal enclosure. We added this information to the text.

L166-167 That sounds dangerous. How is compliance with Health and Safety regulations ensured?

Capillaries and fittings are made from metal and are rated for high pressure applications. Even in the unlikely case of a failure of the capillary tubing, gas could only escape at very low flow due to the small diameter of the capillary tubing. Therefore, the operation is inherently safe.

L171-173 What is the nature of this cold trap and how is its temperature stability ensured? What is its internal volume and how often does it need cleaning/exchanging? What mechanisms are in place to prevent ice building up in the vicinity?

The water trap is cooled by a mixture of dry ice and ethanol. Small changes in temperature are inconsequential as flows from the sample and standard are stabilized later before reaching the MS. The tubing is removed from the dry ice and ethanol mixture after each analysis while dry air flow through the tubing continues. This allows any residual water in the trap to be carried away with the gas stream and any ice build-up on the tubing to melt.

L186 Have any other cycle times been tested?

The cycle time was chosen to ensure complete signal stabilization after switching as determined by evaluating information in Fig. 5. No additional times were tested.

L203-205 If the uncertainty of the correction is 6 per meg, why does it increase the analytical uncertainty by only 2 per meg? Also, what about the uncertainty of 1.5 h measurements?

Uncertainties are added in quadrature here and sqrt(8^2+6^2)=10. By a similar expression, uncertainty of 1.5h measurements would increase to 16 per meg. No changes were made to the manuscript.

L211-213 What are the uncertainties of the corrections?

We thank the reviewer for this question. We added a sentence explaining that "Analytical uncertainty for measurements of $\delta(O_2/N_2)$, $\delta(Ar/N_2)$, and $dX_{CO2}$ is typically better than 1.5 per meg , 11 per meg, and 0.2 ppm (Keeling et al., 1998, 2004), yielding uncertainties of 0.3, 0.11 and 0.2 per meg in the terms $\delta\left(\frac{O_2}{N_2}\right)X_{O_2}$, $\delta(Ar/N_2)X_{Ar}$, and $dX_{CO_2}$ "

L216 I'm not sure that the derivation of this relationship should be part of a Discussion section as it's rather a methodological part of the manuscript. Also, how much larger than relative changes in N2 are relative changes in CO2?

Seasonal changes in atmospheric $N_2$ due to dissolution in the ocean are on the order of a few per meg. Changes in $CO_2$, in contrast, are currently around 2.5 ppm per year or around 6000 per meg. We have moved the derivation into methods and given it its own section.

L217-265 This section focuses on the multitude of possibilities that the new technique might enable, but fails to draw much attention to its limitations (e.g. the large amount of air required, which is not easily acquired from the stratosphere, or the long analysis times, which will limit constraints on spatial gradients in volcanic plumes or near oil or gas facilities).

We thank the reviewer for this suggestion and added a paragraph at the end of the discussion section that summarizes important limitations of our measurement system. Efforts are ongoing to address these limitations, lessen sample size and pressure requirements, and to reduce the duration of individual measurements.

L235 Do the authors perhaps mean "possible complications that affect 3He measurements" here?

A potential problem for estimating a $^4$He source from $^3$He/$^4$He is that $^3$He emissions from different sources such as tritium decay could bias estimates of the $^4$He source. We changed the text to make this clearer.

The second "avoids the need" was removed.

L278-282 I'm not sure that it's necessary to list all these references again. It certainly doesn't help the readability.

Thank you! We removed the references as they are extensively covered in the introduction.

L451-453 Very little detail is provided (in both the figure and the caption) on how the piston system looks and operates. Since this is a crucial part of the inlet system I urge the authors to expand their explanation and perhaps include a cross-section of this chamber. Also, how does the pneumatic control work?

We thank the reviewer for raising this issue. We added more detail to the text and figure on the pistons and pressure stabilization system (see answer to related comment above).

1 sigma. We included the information in the caption.

L490 Table 1 has a lot of empty space and at the same time an enormous amount of footnotes, which make it rather difficult to follow. Consider reorganising, e.g. through naming the references in the table or moving some of the explanations into the caption. Also, what uncertainty range is connected to the assumption of 3He/4He = 3e-8?

We reformatted the table to improve space usage. We have not considered uncertainty about $^3$He/$^4$He of fossil fuels because the uncertainty about the actual $^3$He/$^4$He trend dominates. The estimate of $^3$He/$^4$He = 3e-8 is based on Sano et al (2013).

**Anonymous Referee #2**

*\* The measurement yields the mole fraction of He in the sample gas matrix, normalized relative to the He mole fraction in a gas standard. However, the method is presented in the manuscript to yield the (absolute?) He/N2 ratio of the sample. This is rather confusing, as N2 is NOT part of the measurement (in fact, N2 is removed from the gas sample before MS analysis). The conversion of the measurement result (He mole fraction) to the He/N2 ratio is based on a simple mathematical manipulation and some assumptions about the composition of the gas matrix. This conversion seems trivial and is not related to the measurement technique; and I don't see the need for it. I suggest to avoid this confusion by removing He/N2 (or He/O2 etc.) as far as possible (from the title and most of the text), and to focus on the true nature of the measurement, i.e., on the (relative) He mole fraction.*

We decided to frame the manuscript in terms of $He/N_2$ rather than the helium mole fraction because $He/N_2$ is more readily interpretable in terms of distinct physical processes than the helium mole fraction. Measurements of the helium mole fraction are influenced by both changes in helium and changes in any other component of air. Anthropogenic activity has already led to dramatic changes in the composition of our atmosphere and these changes would obscure the signals from processes that are unique to helium. Because an important contribution of the paper is to develop potential scientific applications of the helium mole fraction measurement, we believe the current framing of the manuscript best serves this purpose. However, we agree that it is critical to clearly distinguish between the helium mole fraction, i.e. the quantity actually measured, and $He/N_2$. To this end, we have altered the title and abstract of the manuscript to avoid confusion.

*\* The manuscript is missing a review/overview of existing techniques for He analysis in air (3He/4He, He/Ne, He mixing ratio). Without this background, it is difficult to understand on what grounds the new method was designed, and how it improves on previous methods, both in terms of instrumental techniques and scientific applications. The manuscript should be revised to better develop the link of the new method to existing techniques. What were the design targets for the new instrument? What was the design approach to achieve these targets? Why was the new method implemented in this way?*

We thank the reviewer for these suggestions. A high precision $^4He/N_2$ measurement has never been attempted before and opens new fields of research. We added a brief explanation of the analytical technique used for helium isotope analysis, and an additional paragraph discussing measurements of the helium mixing ratio. The manuscript in its current form already describes the inherent limitations of the helium isotope method and outlines advantages of a $^4He/N_2$ measurement (lines 45--53). The lack of analytical precision in the helium isotope measurement was a key motivation for developing the $^4He/N_2$ measurement. Potential scientific applications enabled by the new measurement technique are discussed in detail in the introduction of the manuscript as well as the discussion section. We believe a discussion of He/Ne would mostly distract from the main aims and application of the paper.

*\* As it is, I am not convinced about the scientific utility of the new method: (1) Earlier work with static MS systems for He isotope analysis showed that their 0.2% precision was sufficient to resolve the expected atmospheric He variations (for example: Mabry et al, EPSL, 2015). Why exactly is a new method with approx. 100x better precision required to study the evolution of the atmospheric He?*

Recent studies using helium isotopes to constrain trends in atmospheric helium abundance have found no significant trend (Lupton and Evans, 2013; Mabry et al., 2015; Boucher et al., 2018). This is contradicted

by theoretical predictions of a $^4$He build-up in bottom-up approaches (e.g., Oliver et al., 1984; Pierson-Wickmann et al., 2001). A likely explanation for this conundrum is that the anthropogenic signal is too small to be detected with the current analytical precision, which motivates work to improve atmospheric helium measurements. Moreover, atmospheric helium changes in the stratosphere and troposphere from gravitational fractionation are similarly undetectable with current methods, providing a compelling impetus to develop new analytical method.

(2) Most publications on atmospheric He acknowledged that the quality of the historic/archived air samples put major limitations on the uncertainty of reconstructing the atmospheric He abundance (for an overview see for example: Brennwald et al, EPSL, 2013 / reference missing in the manuscript). Further improving the instrumental precision of the He analysis does not help with this fundamental issue. I am therefore not convinced that an improvement of the instrumental precision is very useful to study some of the effects noted by the authors.

Cylinders containing archived air dating back to 1974 are available at Scripps Institution of Oceanography. Similar cylinders are routinely used as standard cylinders for $O_2/N_2$ and $Ar/N_2$ analysis. These cylinders have shown very limited drift over 20 years (Keeling et al., 2007) suggesting that under the right circumstances, archived air should provide a faithful record of atmospheric helium changes over the last 4 decades. Note that Brennwald et al (2013) is not cited in the manuscript as it has been superseded by Mabry et al. (2015)

(3) With an analysis time of 6-8 hours and a sample gas consumption of 28 mL per minute, a single analysis requires about 10-14 L of sample gas. However, typical samples of historic/archived air are typically in the order of 0.1-100 mL (see refs. cited in the manuscript, and as given above), which is 2 to 5 orders of magnitude too low for the new method described in the manuscript. The large gas consumption of the new method therefore severely limits the technical applicability to large-volume gas archives, which are scarce and may not be readily available for consumption to a (destructive!) analysis.

The reviewer raises an important limitation of the analysis method. This and other limitations are now discussed in a separate discussion section. The need for samples of 10-14l is however not prohibitive as the aforementioned high-pressure cylinders still contain about 3000 l (STP) of old air.

In order to illustrate the scientific potential of the new method, the manuscript should be extended with application examples targeted at the scientific questions described in the introduction. I suggest to add a few measurements of real world historic/archived air samples to demonstrate the true utility and suitability of the new method to study the He change due to fossil fuel extraction (and analogous for stratospheric circulation).

Real world measurements of archived air will be presented in a separate paper. To illustrate the fidelity of the measurement, we would like to instead point the reviewer to Cylinder A shown in Fig. 6. Cylinder A was filled in 2008 and is compared to air pumped in 2019 on the same pumping system. The two cylinders indeed demonstrates an increase in $^4$He by about 700-800 per meg over 11 years.

* What is the purpose of the cold traps at the gas inlet system? My guess is that they are meant to remove water vapor from the sample/standard gas streams, but I am not sure. This should be described better.

That is correct and was clarified in the text.

* The intensity of the He ion beam in the MS is controlled by the sample/standard gas flow to the gas inlet system. As far as I can tell, the precision of the analysis result is therefore controlled crucially by the inlet system, and in particular by the cold traps and the pressure regulators: (1) Cold traps: the efficiency of the water removal from the gas streams is likely not stable over time, and will therefore introduce a variation of the He mixing ratio in the gases. How can the cold trap variation be avoided such the He mixing ratio in the gas stream is stable to 10 per meg or better?

We thank the reviewer for this question. The mole fraction measured is defined on a dry air basis, so removal of water is mandatory and does not result in a bias, by definition. The stability of the mass flow to the getter oven and MS is indeed critical to the integrity of the measurement. The mass flow stability is ensured by the combined stability of the thermally insulated crimp and the actively controlled pressure in the stabilization chamber. Any pressure changes caused upstream for example at the water traps are cancelled out. At -80C, the water vapor content of air drops to less than 1 ppm. Even if there where small changes in trap temperature, we would therefore expect these to be << 1ppm and hence not significantly impact the helium mole fraction measurement. Note that we do not use regulators but crimped capillaries to regulate flow from high pressure cylinders to ensure metal seals throughout the system.

(2) The gas pressure at the main capillary inlet directly controls the gas flow into the capillary, and hence the He flow into the MS. Similarly to the cold traps, this seems to imply that the pressure at the capillary inlet must be stabilized to 10 per meg or better. How is this technically possible? I don't have a strong background with large-volume dynamic MS gas analysis, so I may be overlooking something that may be obvious to the authors (I am mostly into static and low-volume dynamic MS). However, I believe it would be very difficult to achieve such tight stability controls over the gas inlet system, and there must be some way to avoid or compensate such variations in the gas inlet in order to achieve the 10 per meg precision in the He mixing ratio of the gas sample. I feel these points need to be explained better.

We appreciate the reviewer's question and expanded the discussion of the stabilization chamber and the sliding tubing in the manuscript. During operation, the pressure in the stabilization chamber is held constant at $96.5 \text{ kPa}$ to within about 0.0133322 Pa. This corresponds to a stability of 0.14 per meg. Such precise pressure control is made possible by differential pressure gauges, which achieve a resolution of 1/10000 of their full range (full range is 0.2 Torr on our gauge).

* Title: "Measurements of ..." seems to indicate that the focus of the paper is to present new measurement data. I'd suggest to change focus to the new analysis method / technique.

We thank the reviewer for the suggestion changed the title to "A method for resolving changes in atmospheric He/N$_2$ as an indicator of fossil fuel extraction and stratospheric circulation".

* Line 26/27: "...gases heavier than air in the stratosphere..." –> what gases other than air are in the stratosphere? Also, does the gravitational separation only apply to heavier gases, not lighter ones?

Gases heavier than air such as argon are depleted in the stratosphere due to gravitational separation while gases lighter than air such as helium are enriched by the same process. We changed the language of the text to clarify this.

* Line 56/57: I'd assume that N2 and other "noble" gases are seasonally variable due to atmosphere/ocean gas exchange, which is subject to the temperature dependence of N2 solubility in the water (e.g., Keeling et al., Tellus 322–338, 2004). Note that the He solubility dependence on temperature is much less than

for N2. While this effect may be relatively small, it should at least be noted in the text (if the N2 normalization is not removed entirely, see above).

The role of ocean heat exchange is indeed small. This is discussed in a full paragraph in the manuscript (lines 102-107).

* Line 83: Using the equations given above, I calculate del(He/N2) = -6.4, not 7.5 (note the sign!). Please check.

We thank the reviewer for this question. Inserting all the terms in the equation, we get a gravitational fractionation signal that is 7.5 times greater and of opposite sign for $He/N_2$ than for $Ar/N_2$. We clarified this important point in the text. Note that the diffusivities used are for the gases in air, not in a binary mixture.

* Line 95: "...no significant trend in atmospheric 3He/4He has been observed..." –> there are many different studies with different (and sometimes contradicting!) conclusions regarding the existence and size of the atmospheric 3He/4He change. Considering the work cited in the manuscript it seems wrong to say "no significant trend was observed", and this argument needs to be revised. To this end, it might be useful to take a look at Brennwald et al, EPSL, 2013, which has a compilation and comparison of different studies on the atmospheric 3He/4He change, and also presents some measurements on He changes observed in archived air samples.

The full sentence this quote is taken from is "However, in contrast to these predictions and some earlier observations (Oliver et al., 1984; Sano et al., 1989, 2010; Pierson-Wickmann et al., 2001), no significant trend in atmospheric $^3He/^4He$ has been observed using archived air samples spanning from the beginning of the 20[th] century to today." It is correct that observations by Sano et al. suggested a trend in atmospheric helium isotopes. This has been disputed by several other studies (e.g., Lupton and Graham, 1991; Lupton and Evans, 2004, 2013; Mabry et al., 2015; Boucher et al., 2018). Of particular significance here is that all 3 studies of direct air samples published in the last 8 years did not find any significant trend. The analysis of Brennwald et al. (2013) was repeated and corrected by Mabry et al. (2015) and is therefore not included in this list. We believe our phrasing adequately reflects the disagreement about observed trends of $^3He/^4He$ and appropriately emphasizes recent results.

* Line 144: "...compared to the 30-second switching timescale." –> what does this refer to? Switching of what?

This refers to the timescale of switching between sample and standard gas in the inlet system. We clarified the wording in the text.

* Line 156: "Background concentrations..." –> what "background" is this? "Blank"? "Residual"? m/z=4 signal with analysis of He free gas? Or with the inlet to the MS closed? Please define.

The background is determined with the inlet closed upstream of the getter oven, but all other aspects of the analytical system in their normal measurement mode (e.g. high voltage ion acceleration and magnetic field strength appropriate for 4He analysis, etc). We modified the manuscript accordingly.

* Line 172: The geometry of the cold trap crucially controls the operation and performance of the cold trap. Is this a U shaped tube? Or a 'washing flask' type? Or something else? Please explain the details of the cold traps ("made from 1/4" stainless steel" is not sufficient).

We are grateful for this excellent question and have added more details to the description of the (U-shaped) trap system.

* Line 175–178: Why is the performance of the getter important (as long as it is not "dead" and works as a pump to draw the air matrix into the capillary)? As far as I can tell, the He flow rate into the MS is identical to the He flow rate into the getter. I therefore don't see how the getter can affect the He analysis (as long as the getter operation is stable between the analysis runs of the sample and standard gases). How does the getter affect the performance of the He analysis? Does the getter performance affect the He analysis at all? Please explain.

An effective removal of $N_2$ and $O_2$ is important to reduce competition with He for electrons in the gas mixture that reaches the MS source. This ensures a brighter He+ beam. We added a sentence to clarify this. It is also important to avoid introducing too much gas into the MS because too high pressures will lead to source instability and ultimately a failure of the instrument.

* Line 201: What does "the 5% level" refer to? Is this a statistical "significance level"? What kind of statistics? Please explain.

This refers to the weighted multiple linear regression analysis described in the text. We changed the phrasing of the manuscript to make this more apparent.

* Fig. 7 shows the measured zero effect, with a substantial scatter (about +/- 50 per meg or so). However, the discussion in the main text indicates a zero-effect uncertainty of only 6 per meg. I am confused because I don't understand how the +/- 50 per meg scatter is consistent with the 6 per meg uncertainty. I believe my confusion is due to some ambiguity in how the zero-effect is quantified and compensated during the analysis and data processing routines. This should be explained better.

We thank the reviewer for this question. Lines 197-202 describe the results of a weighted multiple linear regression analysis of the zero-enrichment test. Uncertainties and statistical significance are products of this established statistical method. We improved the phrasing of the paragraph to explain this better.

* Tab. 2: what is the meaning of "scaled" Xe peak heights? What kind of "scaling" was applied, and how? Please explain.

Xe was not directly observable due to limitations of the dynamic range of a MAT 253 configured to measure He. The MAT 253 has two different magnet ranges, only one of which enables He measurements, and it is prohibitively time consuming to switch between ranges during a mass scan. Instead we report expected Xe peak heights based on previous knowledge about the relative ion yield of Kr and Xe from an air sample. The manuscript was clarified.

**Anonymous Referee #3**

*I think the analytical precisions of mass spectrometry can be classified into "internal precision", "internal reproducibility" and "external reproducibility" (see Bender et al., 1994, Geochim. Cosmochim. Acta.). Internal precision indicates a standard error of xx cycles in a 1-block analysis, and internal reproducibility indicates a standard deviation of the repeated analyses of several blocks. I think the authors did not distinguish between internal precision and internal reproducibility in the paper, and the presented internal precision of ±15 per meg for sample run of 1.5h and ±8 per meg over 6h (line 186-187) are standard error (1sigma/sqrt(n), n = number of the total cycles ~ about 1,800 and 7,200 for 1.5h and 6h, respectively). The internal precision is shown as error bars in Fig. 6, and the variability of every cycle measurements is presented in Fig. 3 (b) (the unit "per meg" is needed for the y-axis label of Fig. 3 (b)). I hope I have not misunderstood or misinterpreted the presentation.*

We thank the reviewer for this suggestion. We have clarified the distinction between the standard error of cycles (here called internal precision) and the standard deviation of repeat analyses (here called external reproducibility) in the manuscript. We now described the nature of the error bar in figure captions. We also corrected the y-axis label of Fig. 3 (b).

*2) Fig. 6 shows d(He/M) values of 8 (5) times repeated analyses of the 6-8h measurements of Cylinder A (B) against La Jolla air. I think it corresponds to the external reproducibility in the context of Bender et al., but I have some concerns: Is the La Jolla air identical to the "He/N2 reference material (line 163-165)", "standard gas cylinder (line 188)", and "reference (Fig. 2)"? Please clarify to avoid confusion. Assuming that the standard and La Jolla air are identical and used as the reference in Fig. 2, how do the authors ensure the short-to-long term stability of the La Jolla air? In other words, how long does it take to obtain the results of 8 or 5 times repeated measurements shown in Fig. 6? If it takes several days, then I agree that the stability of the standard is enough for the period, however, future work is needed to confirm much longer-term stability for the observations of seasonal and interannual variations in the atmospheric He/N2 ratio.*

A high-pressure cylinder containing La Jolla air collected in 2019 has served as the reference gas for this work. The short-term stability of this standard against another high-pressure cylinder is demonstrated in Figure 6. Data for this Figure was collected over a roughly 2-week period. Future work is needed to evaluate the long-term stability of the analysis system as outlined by the reviewer and discussed in the manuscript. We have revised the naming of the reference gas and made it consistent throughout the paper.

*3) Please provide information related to the needed minimum and maximum inner pressures of the sample and reference gas to maintain the appropriate flow rate of 27-28 mL/min using capillary. In this regard, I sort of remember Scripps flask samples being collected at an atmospheric pressure. If so, I don't think these flasks can be used for the He/N2 analysis, as described in the present study. How do the authors collect pressurized air samples without significant fractionation of He and N2?*

The pressure downstream of the water traps and upstream of the pressure stabilization chamber is roughly 3 atm to allow sufficient flow into the chamber when the chamber is kept at 96.9 kPa. The reviewer is correct that Scripps $O_2$ and $CO_2$ flasks are collected at atmospheric pressure, and the new method we describe here does not involve the Scripps $O_2$ and $CO_2$ flasks. The pressurized samples used

here are pumped with a heavy-duty compressor and fractionation is largely avoided by the quantitative capture of a very large amount of air (~3000 liters) in a short time with minimal leakage.

Some changes will be needed to use Scripps $O_2$ network flask directly as samples. Compatibility might be achievable by simultaneously (i) dropping the pressure in the stabilization chamber, (ii) adjusting the flow restriction after the stabilization chamber to offset the reduction in flow to the MS, and (iii) increasing the conductance of tubing delivering gas into the chamber. None of these changes should present a major obstacle and work is ongoing to accommodate flask samples. In the meantime, the system can accept pressurized air samples including a suite of high-pressure cylinders available at SIO that contain air sampled over the last 40 years. Acceptable agreement in $O_2/N_2$ and $Ar/N_2$ between flask samples and high-pressure cylinders routinely filled at SIO suggests they might be suitable to reconstruct atmospheric changes in $He/N_2$. Results from these cylinders will be presented in a future publication.

4) Related to the comment 3) above, there does not seem to be any discussion of the analyses of the locally pumped ambient air shown in Fig. 2. I would be very much interested in hearing from the authors some information regarding the ambient air analyses, since it will serve as an indicative evaluation of "true" external reproducibility, including air sampling procedures, such as inlet fractionation, and leakage or permeation of He through pump diaphragm.

The measurement system was initially setup for analysis of ambient air but unfortunately never deployed because of limited instrument time. We removed it from the text and diagram as there is no data available to evaluate performance.

5) I think the response of d(He/N2) measured by the mass spectrometer to the actual H2/N2 ratio of a sample air will likely be linear, but some checks are needed. Please provide some concrete evidence to that effect.

We thank the reviewer for this question. On the MAT 253, ion beams are reported in voltages rather than currents for technical reasons. The ion beam current in counts per second (I) is related to the voltage (V) read by the resistance (R) according to Ohms' law:        I(Cps) = V/R

The manufacturer's instrument specifications state that nonlinearity of the MAT253 mass spectrometer for $CO_2$ isotopes is better than 0.06 per mille per 1 Volt change in the ion beam when introducing pure $CO_2$ into the MS. The $^4$He beam is collected on a $10^{12}\Omega$ resistor and roughly yields a voltage of 1.5V. The largest $^4$He beam strength difference between sample and standard that we expect to measure is about 3 per mille or 0.0045V. If the $^4$He beam behaved like $CO_2$ isotopes in the MS the nonlinearity bias would be 4.5×0.06=0.00027 per mille.

However, a leading cause for non-linearity in a MS are changes in total source pressure, which are minimal in our application. As the gas mixture that enters the MS source consists mostly of Ar and atmospheric Ar concentrations are stable to within a few per meg, we expect the MS response in our application to be absolutely linear.

6) The potential application of He/N2 to evaluate interannual variability in the stratospheric circulation (±375 per meg/yr, expected value) is very interesting and is an excellent idea, considering that it can provide better signal-to-noise ratio than Ar/N2. In order to extract the gravitational fractionation signal of He/N2, it is necessary to subtract the He/N2 change due to chemical processes such as fossil fuel extraction (which yields 35 – 350 per meg/yr of secular He/N2 trend at the surface) from the observational

results in the stratosphere. I guess the authors have plans to observe surface He/N2 variations, so that a precise secular trend in the troposphere will be achieved in the near future. However, as reported by Engel et el. (2009 Nature Geoscience), CO2 or SF6 age of the stratospheric air samples show interannual variability by about ±1 years. This interannual variability of the age corresponds to ±35 – ±350 per meg of the stratospheric He/N2 change due to the fossil fuel extraction, if we ignore attenuation of the interannual variability from the surface to the stratosphere. Therefore, to evaluate the interannual variability in the stratospheric circulation based on gravitational fractionation of He/N2, precise determination of He/N2 age will be important. However, as Ray et al. (2017 JGR) and Sugawara et al. (2018 ACP) reported, CO2 age and SF6 age do not necessarily agree with each other. Given these issues, I would be very much interested in hearing how the authors are planning to determine the age of He/N2.

We thank the reviewer for the excellent question. As explained by the reviewer, a correction for the tropospheric trend will be needed for the calculation of age of air (AoA) from stratospheric helium measurements. Furthermore, a fundamental relationship between gravitational fractionation of He/N2 and AoA in the stratosphere will need to be determined. This could be done for example by building on previous modelling work by Birner et al. (2020). The atmospheric transport model TOMCAT has been validated for the relationship between $Ar/N_2$ and AoA in the lower stratosphere and results for He/N2 could be obtained by a simple scaling or implementation of He as a new tracer in the model. These modeling results would establish a $He/N_2$-AoA relationship that would be valid in the absence of any tropospheric $He/N_2$ trend. However, a correction for any potential tropospheric helium variability would be needed to apply this calibration to real data. Depending on the complexity of the tropospheric helium variability, different approaches are potentially feasibility. In the simplest and most likely case of a linear (or at least near-linear) increase of tropospheric $He/N_2$, the $He/N_2$-AoA relationship could be modified by subtracting the linear $He/N_2$ change. In the case of a more complex tropospheric history, the tropospheric record would have to be convolved with an assumed age spectrum to obtain an appropriate correction. This is possible but will rely on assumptions about the shape of the age spectrum. We believe this goes beyond the scope of the paper and is therefore not addressed in the text.

7) Line 198: "9 psi, 14 psi, and 16 psi. . ." should be changed to "9, 14, and 16 psi".

The repeated usage of the unit was removed.

8) Line 202: "-9.61±7.2 per meg, 1±3.7 per meg, and -15.7 per meg. . ." should be changed to "-9.61±7.2, 1±3.7, and -15.7 per meg. . .".

The repeated usage of the unit was removed.

9) References: Please change "2" in CO2, O2/N2, Ar/N2 and SF6 to subscripts.

We thank the reviewer for catching this oversight. We implemented correct usage of subscripts in the reference section.

[revised manuscript text omitted]